

# A model test study on the parameters affecting the cyclic lateral response of monopile foundations for offshore wind turbines embedded in non-cohesive soils.

Dennis Frick[1], Martin Achmus[1]

[1]Institute for Geotechnical Engineering, Leibniz University Hannover, Hannover, 30167, Germany

*Correspondence to*: Dennis Frick (frick@igth.uni-hannover.de)

**Abstract**

During their service life, monopiles supporting offshore wind turbines are subjected to large numbers of lateral cyclic loads resulting from complex environmental conditions such as wind and waves varying in amplitude, direction, load eccentricity
and frequency. The consequential accumulation of displacements and rotations of the foundation structure with cyclic loading is one key concern in the design of monopiles. Nevertheless, the relevant offshore guidelines do not provide suitable procedures for predicting such deformations. Although there are several methods for this purpose in literature, some of them produce very different or even contradictory results, which prevents a consistent approach to dimensioning. This paper briefly summarizes the current standardization regarding design of monopiles for cyclic lateral loading and provides some examples of possible
prediction models from the literature. To highlight the need for further research, the predictions according to different approaches are compared and evaluated by a calculation example and a parameter study. Further, the results of a small-scale 1g model test campaign on the load-displacement behaviour of monopile foundations subjected to lateral cyclic loading and the influencing parameters are presented, evaluated and compared with the findings of other research groups. In this way the tests results can help to support or improve model development and provide insight into key issues relevant to monopile design.
The parameters that have been assessed include the cyclic load magnitude, cyclic load ratio, load eccentricity, soil relative density, the grain size distribution of the non-cohesive bedding material as well as the pile embedment length.

## 1 Introduction

Offshore wind energy is one of the promising solutions for sustainable energy, but for the wind industry to be competitive, it is vital that costs are significantly reduced for future projects. This can be achieved on the one hand by introducing new
technologies and on the other hand by improving existing technologies and design methods. One of the areas where costs can be reduced is the support structure, which accounts typically for about 16 to 35% of the total cost of an offshore wind turbine (OWT) and whose cost increase substantially with water depth (Bhattacharya et al., 2021). With regard to continuously increasing water depths of future wind farm sites and projects, an improved design of the foundation structure can therefore make a significant contribute to the competitiveness of offshore wind energy.

Up to now, the prevailing support structure for offshore wind energy converters at low to medium water depths is the monopile foundation, a single pile with large diameter (D) and a relatively small ratio of embedment length (L) to the diameter (L/D), that transfers the predominantly horizontal loads from the action of wind and waves into the seabed. Its popularity can be addressed to its suitability for mass-fabrication, robustness for most soil conditions, a relatively simple design and therefore cost efficiency. To extend the range of application of the monopile and make use of the related benefits, pile diameters have
to be extended (leading to decreasing L/D-ratios) and more accurate design methods, specifically tailored to the offshore wind industry have to be developed.

A governing factor in the design of monopile foundations is compliance with serviceability limit state and associated strict tilting tolerances. This means that an accurate prediction of pile displacement and rotation accumulation resulting from cyclic occurring horizontal loads plays a key role for the final dimensions of the foundation structure and therefore its costs. However,
current offshore guidelines do not provide appropriate procedures for the prediction of pile displacement accumulations, which is especially true for monopiles, which due to their large dimensions and low L/D-ratios have a significantly different load-displacement behaviour than long and slender piles. For this reason, a variety of new empirical and numerical approaches for



the estimation of cyclic deformation behaviour of monopiles have been proposed in literature. Although these methods were usually developed specifically for monopile foundations, they sometimes provide very different and partly contradictory results

with respect to the resulting deformations and the governing parameters.

The article at hand first summarises the current standards and developments regarding the estimation of permanent deformations of offshore monopiles, before selected prediction models are compared with each other on the basis of a calculation example and a parameter study. In order to gain further insight into the deformation behaviour of monopiles due to lateral cyclic loading, results of a comprehensive test campaign of small-scale 1g model experiments are presented and

discussed. The results are used to identify the governing parameters and to evaluate existing empirical approaches. Based on the results, qualitative conclusions can be drawn. The findings can thus contribute to a better understanding of the complex processes associated with the cyclic load-bearing behaviour of piles and to the development of improved calculation approaches.

## 2 State of the art

The design of offshore structures, such as monopile foundations for offshore wind turbines, is usually based on the latest version of the offshore guidelines, e.g. DNV GL (2018) or API (2014). These also regulate the required design checks, which include the proofs for ultimate limit states (ULS), serviceability limit states (SLS), fatigue limit states (FLS) and accidental limit states (ALS). In case of large-diameter monopiles for OWTs, the SLS proof for long-term lateral cyclic loading resulting from millions of wind and wave loads is often decisive for the dimensioning of the foundation. For this proof, limit values for

permanent pile head displacement or rotation at seabed level are usually specified by turbine manufacturers or structural designers, whose compliance is to ensure the safe and smooth operation of the turbine until the end of its planned service life. As an example, the DNV GL (2018) guideline for this proof provides the usual limit values of 0.5° pile head rotation, including an installation tolerance of 0.25°, which means that the additional accumulated rotation due to lateral cyclic loading must be limited to less than 0.25° in this case. Both the DNV GL (2018) and the API (2014) guideline mention the so-called p-y method

as a possibility to model the pile-soil interaction in horizontal direction and to predict the load-deformation behaviour of a pile foundation due to lateral loads. This method models the pile-soil system by discretising the pile into a number of elastic beam elements, interconnected by nodal points, and with uncoupled soil support springs laterally attached to these nodal points. Loads such as horizontal forces or moments are applied directly to the pile head. The characteristics of the springs (p-y curves) are hereby non-linear and describe the relationship between soil's bedding resistance (p) and lateral pile displacement (y).

Therefore, the reliability of the calculated prediction of pile deformations by this method strongly depends on the chosen formulation of the p-y curves. While API (2014) refers to API 2GEO (2014) for an approach to the construction of p-y curves, the DNV GL (2018) does not recommend a specific approach, but points out that any p-y method to be used for piles larger than 1.0 m in diameter should be validated by means of other methods (e.g. finite element calculations). This remark results from the fact that most p-y approaches (including the API 2GEO (2014) method) are largely based on some well documented

field tests on small-diameter, long and therefore slender piles reported by Reese et al. (1974), Murchinson & O'Neill (1984) and others. Since the load transfer behaviour of such slender and thus more flexible piles with large L/D-ratios differs significantly from that of rigid piles (e.g. typical large diameter monopiles), these methods should not be used for this field of application without further validation. In addition to this general issue regarding the p-y method or most of the approaches for the determination of p-y curves, the DNV GL (2018) guideline states, that the SLS proof of a monopile requires that it

represents the behaviour of the soil under cyclic loading in such a way that cumulative deformations in the soil are appropriately calculated as a function of the number of load cycles at each load amplitude in the applied history of SLS loads. However, no specific procedure for this purpose is mentioned in DNV GL (2018) either. In contrast, the p-y method according to API 2GEO (2014) allows the consideration of cyclic loads by a simple adjustment of the proposed p-y curves by an empirical calibration





factor. When being applied, this factor leads to an overall softer foundation response as well as a reduced pile capacity without
considering the number of applied load cycles, load magnitude or other parameters of the load or the pile-soil system. As the
calibration factor according to the API 2GEO (2014) approach was derived from field tests with less than 100 load cycles in
most cases, this method is widely deemed to be unsuitable for SLS verifications of monopiles for OWTs, especially when
large diameter piles are used. In this context, the API (2014) states that the methods referred to are only intended as
recommendation. Therefore, if further detailed information from advanced soil testing, pile testing in the centrifuge, at model-
scale or even at full-scale are available, then also others methods may be justified.

In summary, it can be seen from the above, both offshore guidelines, while regulating the principles of design of offshore
foundations, do not provide a generally applicable method for pile deformation assessment due to lateral cyclic loading for
SLS verification of large diameter monopiles. Instead, it is up to the designer to choose a suitable method for this purpose.
Accordingly, there are several publications on the subject of cyclic laterally loaded piles in the literature and on how
deformations due to such loads can be predicted. Most of the methods proposed are based either on a limited number of small-
scale model tests at 1g or in the centrifuge, with a few approaches also based on field experiments. Mostly, these approaches
were derived from best fit curves, where it has been found that for a given load level and type of loading, the ratio of the pile
head displacement accumulated after N load cycles ($y_N$) and the maximum displacement reached within the first cycle ($y_1$) can
most generally be described as a function of the number of load cycles (N) by either a power or a logarithmic function as
shown in Eq. (1) and Eq. (2):

$$\frac{y_N}{y_1} = N^\alpha \tag{1}$$

$$\frac{y_N}{y_1} = (1 + t \cdot \ln N) \tag{2}$$

Here, $\alpha$ and t are referred to as accumulation parameters and may be defined differently depending on the chosen approach
taken from literature. It should be noted that according to some methods, also pile head rotations ($\theta_N$, $\theta_1$) are used as
deformation variables in Eq. 1 or Eq. 2 instead of the pile head displacements ($y_N$, $y_1$). The maximum deformation reached
within the first load cycle ($y_1$ or $\theta_1$) is usually determined from monotonic load-displacement or –rotation curves, which in
turn can be calculated using a suitable method as for example finite element calculations, an appropriate p-y method (e.g.
Kallehave et al., 2012; Sørensen, 2012; Thieken et al., 2015) or the PISA-method (see, e.g. Byrne et al., 2017; Byrne et al.,
2019; amongst other). Although both equations (Eq. (1) and Eq. (2)) are often considered to describe the variation of
accumulated pile deformations with number of load cycles, most studies indicate that for a pile-soil system that behaves almost
rigid, the power function according to Eq. (1) gives more accurate results, whereas the logarithmic function better fits a flexible
pile behaviour when subjected to cyclic loading (see, e.g. Peralta, 2010).

While early publications on the topic of cyclic laterally loaded pile foundations focused primarily on the behaviour of long
and slender piles with a limited number of mostly one-way load cycles (see, e.g. Hettler, 1981; Little & Briaud, 1988; Long &
Vanneste, 1994; Lin & Liao, 1999), the interest of the last decade has been mainly in predicting the behaviour of piles with
dimensions and loading conditions typical for offshore monopile foundations (e.g. rigid pile behaviour, higher number of load
cycles, one- and two-way loading). In order to clearly describe constant cyclic loading conditions, the two load parameters $\zeta_b$
and $\zeta_c$ defined by Eq. (3) and Eq. (4) are well established.

$$\zeta_b = \frac{H_{max}}{H_{ref}} = \frac{M_{max}}{M_{ref}} \tag{3}$$

$$\zeta_c = \frac{H_{min}}{H_{max}} = \frac{M_{min}}{M_{max}} \tag{4}$$

In these equations, the reference horizontal force or moment ($H_{ref}$ or $M_{ref}$) are those corresponding to monotonic loading of a
pile soil system at failure or at a reference displacement or rotation ($y_{ref}$ or $\theta_{ref}$) at soil surface. As a geotechnical failure of a





rigid, laterally loaded pile in sand due to monotonic loading can require large pile deformations, it has become common practice to define $H_{ref}$ or $M_{ref}$ not at pile failure, but at significantly lower reference values for $y_{ref}$ or $\theta_{ref}$. Further, $H_{min}$ and $H_{max}$ are the

minimum and maximum horizontal forces being applied to the pile within a load cycle with associated moments $M_{min}$ and $M_{max}$ acting on the pile head at ground level. Therefore, $\zeta_b$ can be interpreted as the cyclic load magnitude, while $\zeta_c$ is the loading type with $H_{max}$ and $H_{min}$ taking positive and negative values for two-way loading, respectively.

In order to investigate the load-bearing behaviour of large-diameter piles in sand subjected to long-term lateral cyclic loading, Peralta (2010) conducted a series of 34 scaled 1g model tests (13 monotonic and 21 cyclic) on model piles (D = 60 mm) with

L/D-ratios within the range of 3.33-8.33 and up to 10,000 load cycles. The tests involved cyclic one-way loading only, with loads being applied with an eccentricity (h) of 240 mm (distance between load application point and soil surface). Both rigid and flexible pile-soil systems with different relative soil densities ($D_r$) and pile bending stiffnesses ($E_pI_p$) were investigated. In addition, also the influence of the cyclic load magnitude $\zeta_b$ was considered. As a result, it has been found that the measured pile displacement accumulations of the rigid pile-soil systems followed a power function as shown in Eq. (1), while a

logarithmic trend (Eq. (2)) was observed for the piles with a more flexible behaviour. For the accumulation parameters given in Eq. (1) and Eq. (2), Peralta (2010) suggests values of $\alpha_P = 0.12$ and $t_P = 0.21$ for rigid and flexible piles-soil systems, respectively, regardless of the soil relative density. An influence of the load magnitude ($\zeta_b$) on the accumulation parameters ($\alpha_P$ and $t_P$) was also not observed; the load magnitude ($\zeta_b$) and soil relative density ($D_r$) correlated only with the value of $y_1$.

LeBlanc et al. (2010) also conducted a series of 21 small-scale model tests (6 monotonic and 15 cyclic) at 1g, in which the

influence of not only the load magnitude $\zeta_b$ but also the loading type $\zeta_c$ and the soil relative density ($D_r$) was investigated in more detail. The rigid model pile had a diameter (D) of 80 mm and a L/D-ratio of 4.5, which is typical for large-diameter monopiles. Lateral loads have been applied with up to 65,370 load cycles (at least 7,400) and an eccentricity (h) of 430 mm resulting in an h/D-ratio of 5.38. In order to take scaling effects into account and to ensure comparability of the dilatancy and shearing behaviour of the soil (dry silica sand) between the model and true scale, the model tests were carried out at relative

densities ($D_r$) of only 0.04 (very loose) and 0.38 (medium-dense). As the shear parameters of the soil are stress-dependent (at least for very small vertical stresses), LeBlanc et al. (2010) provide a graphical relationship between vertical effective stress with reference stress taken at a depth (z) of 0.8 L, soil relative density ($D_r$) and peak friction angle ($\varphi'$), which can be used to convert the relative densities used in the model tests to a true-scale monopile. Based on the results of the conducted cyclic tests LeBlanc et al. (2010) propose the power function approach given in Eq. (5) to describe permanent increases in pile head

rotation ($\Delta\theta$) with load cycle number (N).

$$\Delta\theta(N) = \theta_N - \theta_1 = \theta_1 \cdot T_{b,LB}(\zeta_b, D_r) \cdot T_{c,LB}(\zeta_c) \cdot N^{\alpha_{\theta,LB}} \tag{5}$$

For the accumulation parameter ($\alpha_{\theta,LB}$) they recommend a value of 0.31. The factors $T_{b,LB}$ and $T_{c,LB}$ were identified to be dependent on load characteristics and soil relative density and have been defined in terms of graphical functions (see LeBlanc et al., 2010). While $T_{b,LB}$ increases linearly with load magnitude ($\zeta_b$) and takes larger values for a higher relative density ($D_r$),

the $T_{c,LB}$-function indicating the influence of the loading type ($\zeta_c$) on the pile head rotation accumulation is according to the results of LeBlanc et al. (2010) not affected by soil relative density ($D_r$) and shows a maximum of approximately 4 at a cyclic load ratio of $\zeta_c = -0.6$ (asymmetric two-way loading).

Another approach was proposed by Klinkvort & Hededal (2013), who in their centrifuge tests (5 monotonic and 12 cyclic) on almost rigid model piles with diameter (D) of 28 mm and 40 mm, respectively, and a constant L/D-ratio of 6, applied up to

10,000 load cycles (however, most of the tests involved only 500 load cycles) with a normalized load eccentricity (h/D) of 15. The soil relative density ($D_r$) in these tests ranged from 0.79 to 0.96 and the applied cyclic loads also varied in both their load magnitude ($\zeta_b$) and cyclic load ratio ($\zeta_c$). The results regarding the pile head displacement have been found to follow a power law according to Eq. (1), but unlike to the findings of Peralta (2010), Klinkvort and Hededal (2013) found an influence of the load magnitude ($\zeta_b$) and cyclic load ratio ($\zeta_c$) on the accumulation parameter. An impact of the soil relative density ($D_r$),





however, could not be determined. As a result, Klinkvort and Hededal (2013) defined the accumulation parameter (α) from Eq. (1) as follows:

$$\alpha_{K\&H} = T_{b,K\&H}(\zeta_b) \cdot T_{c,K\&H}(\zeta_c) \qquad (6)$$

Where $T_{b,K\&H}$ and $T_{c,K\&H}$ were defined by two functions depending on $\zeta_b$ and $\zeta_c$, respectively. While the $T_{b,K\&H}$-function indicates a linear increase with cyclic load magnitude ($\zeta_b$), $T_{c,K\&H}(\zeta_c)$ is given by a third-order polynomial with a maximum

value slightly larger than 1 for $\zeta_c = -0.01$ and taking even negative values of up to -1.95 for perfect two-way loading ($\zeta_c = -1$), which means that the accumulation of displacement is reversed for this loading condition and the pile moves back to its initial position.

Li et al. (2015) conducted one of the few field test studies, where four open-ended steel pipe piles with outer diameter (D) of 340 mm, wall thickness (t) of 14 mm and an embedment length (L) of 2,200 mm (L/D = 6.47) were tested in an over-

consolidated fine sand with a relative density ($D_r$) close to 1. All loads have been applied with a normalized eccentricity (h/D) of 1.18. Two pile tests were performed to derive monotonic load- displacement curves and determine $H_{ref}$ (see. Eq. (3)). In the other two tests, cyclic one-way loads ($\zeta_c = 0$) were applied in three load packages of different cyclic load magnitudes, increasing from small to larger values of $\zeta_b$, and with different numbers of load cycles ranging from N = 40 to N = 4,007 for each load package. After cycling, monotonic tests were performed in order to determine the post-cyclic load-displacement response of

the piles and to see the effect of the cyclic loading history. The results of the cyclic tests have been fitted by both power and logarithmic functions as given in Eq. (1) and Eq. (2) and with respect to pile head displacement (y) as well as pile head rotation (θ). Further, a Miner-rule based superposition method with both models (the logarithmic and power law functions) was applied to the results to proof the validity of this method to predict the accumulated pile head response to multi-amplitude lateral cyclic loading. From the evaluation of the results, Li et al. (2015) propose $\alpha_{y,L} = 0.085$ and $\alpha_{\theta,L} = 0.060$ as power law accumulation

parameters for displacement (y) and rotation (θ), respectively. For the corresponding logarithmic accumulation parameters they suggest $t_{y,L} = 0.125$ and $t_{\theta,L} = 0.080$. However, since these values are based on only two cyclic tests with one-way loading, no conclusions can be drawn about the influence of varying soil relative density ($D_r$) or other cyclic load ratios ($\zeta_c$). Regarding the superposition model, Li et al. (2015) found a very good overall prediction of the model with both logarithmic and power functions.

A study involving considerably more model tests and different boundary conditions was conducted by Truong et al. (2019). In this study, 17 centrifuge tests (3 static and 14 cyclic) with different soil relative densities ($D_r = 0.57$ to $D_r = 0.95$), pile slenderness ratios (L/D = 6 and L/D = 11.4) and load magnitudes ($\zeta_b$) were conducted. The model piles had diameters (D) of 11 mm as well as 40 mm. Also normalized load eccentricity was varied between h/D = 2 and h/D = 3 and cyclic loads have been applied with load cycle numbers (N) between 50 and 1,500 with different cyclic load ratios ($\zeta_c$). In addition to these

centrifuge tests, Truong et al. (2019) also considered the test results of Klinkvort and Hededal (2013), Li et al. (2015) and Rosquoët et al. (2007) to develop a new method for the estimation of pile head displacement accumulations with load cycle number (N). Based on the before mentioned results, they suggest a power law as given in Eq. (1) in combination with an accumulation parameter (α) according to Eq. (7) to account for different soil relative densities ($D_r > 0.5$) and cyclic load ratios ($\zeta_c$).

$$\alpha_{y,T} = (0.3 - 0.22 \cdot D_r) \cdot \left[ 1.2 \cdot \left( 1 - \zeta_c^{\,2} \right) \cdot (1 - 0.3 \cdot \zeta_c) \right] \qquad (7)$$

Following this approach, maximum pile head accumulations result from cyclic load ratios of about $\zeta_c = -0.5$ and lower soil relative densities ($D_r$). Further, Truong et al. (2019) could not confirm a significant effect of the load magnitude ($\zeta_b$) or the eccentricity of applied loads (h), although these variables are of course included in Eq. (1) as the displacement due to initial loading ($y_1$) depends on them.

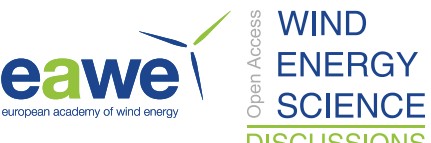

A recent proposal for an approach to calculate the accumulation parameter (α) given in Eq. (1) for pile head displacement from Li et al. (2020) is based on a series of 20 centrifuge tests (2 monotonic and 18 cyclic) on model piles with diameter (D) of 18 mm and an embedment length (L) of 90 mm (L/D = 5). In this study, two different soil relative densities ($D_r$ = 0.5 and $D_r$ = 0.8) have been tested and cyclic loads have been applied with load cycle numbers (N) ranging from 42 to 153, a normalized load eccentricity (h/D) of 8 and several different load magnitudes ($\zeta_b$) as well as cyclic load ratios ($\zeta_c$). Together

with Eq. (1), for the accumulation parameter (α) Li et al. (2020) suggest the formulation of Klinkvort and Hededal (2013) given in Eq. (6), but define a new set of functions to derive $T_b$ and $T_c$ (here referred to as $T_{b,L}$ and $T_{c,L}$). In contrast to Klinkvort and Hededal (2013), Li et al. (2020) found $T_b$ independent from load magnitude to be a constant taking a value of $T_{b,L}$ = 0.07335. The parameter $T_{c,L}$ is given by two equations, each dependent on the cyclic load ratio ($\zeta_c$), for the two soil relative densities ($D_r$) used in their test series. According to this approach, the largest accumulation parameter (α) and therefore

most displacement accumulation results for asymmetric cyclic two-way loading with $\zeta_c \approx$ -0.3 and a soil relative density ($D_r$) of 0.5.

From the above, it can be seen that a variety of different approaches exists for the estimation of cyclic deformation accumulations. Even if the studies and methods mentioned here represent only part of the approaches to be found in the literature, it is already apparent from this, that due to the complexity of the mechanisms driving displacement accumulation

and inherent differences in reported tests, disagreements in results of different studies and the approaches derived from them are to be expected. Therefore, careful examination of the assumptions and the applicability of each proposed method is required. To facilitate a comparison of the individual methods mentioned and the most important underlying boundary conditions, these are summarized briefly in Table 1.

**Table 1: Overview of models for cyclic displacement or rotation accumulation resulting from lateral cyclic loading**

| Reference | Model | Test type | Load cycles | Pile-soil-system | Model parameters |
|---|---|---|---|---|---|
| Peralta (2010) | $y_N = y_1 \cdot N^{\alpha_P}$ <br> $y_N = y_1 \cdot (1 + t_P \cdot \ln N)$ | 1g <br> 13 static <br> 21 cyclic | 10.000 | D = 6 cm <br> L/D = 3.33-8.33 <br> h = 240 mm = const. <br> $D_r$ = 0.45 & 0.65 | Rigid piles: $\alpha_P$ = 0.12 <br> Flexible piles: $t_P$ = 0.21 |
| LeBlanc et al. (2010) | $\theta_N = \theta_1 \cdot \left(1 + T_{b,LB} \cdot T_{c,LB} \cdot N^{\alpha_{\theta,LB}}\right)$ | 1g <br> 6 static <br> 15 cyclic | 7.400– 65.370 | D = 8 cm <br> L/D = 4.5 = const. <br> h/D = 5.38 = const. <br> $D_r$ = 0.04 & 0.38 | $\alpha_{LB}$ = 0.31 <br> $T_{b,LB}(\zeta_b, D_r = 0.38) = 0.414\, \zeta_b - 0.023$ * <br> $T_{b,LB}(\zeta_b, D_r = 0.04) = 0.303\, \zeta_b - 0.044$ <br> $T_{c,LB}(\zeta_c) = a\, \zeta_c^4 + b\, \zeta_c^3 + c\, \zeta_c^2 + d\, \zeta_c + e$ ** |
| Klinkvort & Hededal (2013) | $y_N = y_1 \cdot N^{T_{b,K\&H} \cdot T_{c,K\&H}}$ | Centrifuge <br> 5 static <br> 12 cyclic | 250-10.000 | D = 2.8 cm & 4.0 cm <br> L/D = 6 = const. <br> h/D = 15 = const. <br> $D_r$ = 0.79 - 0.96 | $T_{b,K\&H}(\zeta_b) = 0.61\, \zeta_b - 0.013$ <br> $T_{c,K\&H}(\zeta_c) = (\zeta_c + 0.63)\,(\zeta_c - 1)\,(\zeta_c - 1.64)$ |
| Li et al. (2015) | $y_N = y_1 \cdot N^{\alpha_{y,L}}$ <br> $y_N = y_1 \cdot (1 + t_{y,L} \cdot \ln N)$ <br> $\theta_N = \theta_1 \cdot N^{\alpha_{\theta,L}}$ <br> $\theta_N = \theta_1 \cdot (1 + t_{\theta,L} \cdot \ln N)$ | Field tests <br> 2 static <br> 2 cyclic | 3.173-5.017 | D = 34 cm <br> L/D = 6.47 = const. <br> h/D = 1.18 = const. <br> $D_r \approx$ 1.0 | $\alpha_{y,L}$ = 0.085 <br> $t_{y,L}$ = 0.125 <br> $\alpha_{\theta,L}$ = 0.060 <br> $t_{\theta,L}$ = 0.080 |
| Truong et al. (2019) | $y_N = y_1 \cdot N^{\alpha_{y,T}}$ | Centrifuge <br> 3 static <br> 14 cyclic | 50-1.500 | D = 1.1 cm & 4 cm <br> L/D = 11.4 & 6 <br> h/D = 2 & 3 <br> $D_r$ = 0.57 - 0.95 | $\alpha_{y,T} = (0.3 - 0.22 D_r)\left[1.2\left(1 - \zeta_c^2\right)(1 - 0.3\zeta_c)\right]$ |
| Li et al. (2020) | $y_N = y_1 \cdot N^{T_{b,L} \cdot T_{c,L}}$ | Centrifuge <br> 2 static <br> 18 cyclic | 42-153 | D = 1.8 cm <br> L/D = 5 = const. <br> h/D = 8 <br> $D_r$ = 0.5 & $D_r$ = 0.8 | $T_{b,L}(\zeta_b) = 0.07335$ <br> $T_{c,L}(\zeta_c, D_r = 0.8) = -1.707(\zeta_c + 0.31)^2 + 0.949$ <br> $T_{c,L}(\zeta_c, D_r = 0.5) = -1.14(\zeta_c + 0.323)^2 + 1.263$ |

*$T_b$ and $T_c$ functions fitted based on the graphical representations given in LeBlanc et al. (2010)

** Polynomial factors for the determination of: $T_c$ ($\zeta_c \leq$ -0.3): a = 113.33; b = 288.56; c = 238.88; d = 73.48; e = 9.94

$T_c$ ($\zeta_c >$ -0.3): a = 3.06; b = -6.50; c = 5.22; d = -2.76; e = 0.99





### 3 Comparison of different empirical approaches for the estimation of cyclic lateral deformation


To allow not only a qualitative but a quantitative comparison of the different empirical methods for the prediction of monopile deformation accumulations resulting from lateral cyclic loading given in Table 1, a calculation example and a parameter study are presented in the following. Since all the above approaches describe deformation of the pile accumulated after a certain number of load cycles ($y_N$ or $\theta_N$) as a function of the initial deformation after first loading ($y_1$ or $\theta_1$), a monotonic load-

displacement or load-rotation curve, respectively, is the basis for further calculations. Therefore, Fig. 1 shows the response of a steel pile ($E_p = 21 \cdot 10^7$ kN/m², $\gamma_s = 68$ kN/m³, $\nu_s = 0.27$) with typical monopile dimensions (D = 8 m, t = 0.08 m, L = 32 m) and an L/D-ratio of 4 to monotonic loading. It was calculated for a load eccentricity (h) of 32 m with the p-y method proposed by Thieken et al. (2015) using the freely accessible pile design program IGTHPile V3.1 (Terceros et al., 2015). The relevant soil parameters for the calculation representative of a homogeneous and dense sand layer are given in the bottom line of

Table 2.

On the one hand, these curves can be used to determine the displacement or rotation ($y_1$ or $\theta_1$) for a given load, and on the other hand they can be used to determine the reference load ($H_{ref}$) for the definition of the load magnitude ($\zeta_b$) according to Eq. (3). However, as there is no single criterion for determining $H_{ref}$, this value had to be evaluated for each approach according to the specifications of the respective authors. Relevant deformation criteria for the definition of the reference load ($H_{ref}$) and

corresponding values taken from Fig. 1 are given in Table 2. It should be mentioned that due to the different specifications regarding the reference load ($H_{ref}$), a direct comparison of load magnitudes ($\zeta_b$) between different approaches (see Table 1) is not possible. In order to be able to make a direct comparison of the various prediction models within the framework of the parameter study, the cyclic loads were defined in terms of absolute magnitude ($H_{max}$) rather than their relative load magnitude ($\zeta_b$). Since the relative load magnitude ($\zeta_b$) is nevertheless required as an input variable for some of the models shown in

Table 1, it was determined and summarized in Table 2 for a bandwidth of horizontal loads ($H_{max}$) depending on the associated value of the reference force ($H_{ref}$) for each method. Here, it can be seen that both the reference pile capacities ($H_{ref}$) and therefore also the associated relative load magnitudes ($\zeta_b(H_{max})$) vary over a wide range depending on the chosen criterion, even exceeding the value of 1 when $\theta_{ref} = 0.5°$ is applied as proposed by Truong et al (2019).

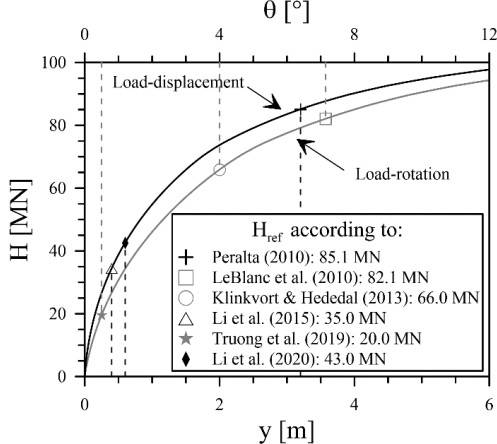

**Figure 1: Monotonic load-deformation response of a horizontally loaded pile (D = 8 m, t = 0.08 m, L = 32 m, h = 32 m) in dense sand calculated with the p-y method according to Thieken et al. (2015) and reference loads ($H_{ref}$) defined by different criteria (see Table 2).**






**Table 2: Different criteria for the definition of the reference horizontal load ($H_{ref}$) and corresponding values determined from Figure 1 with relative load magnitudes ($\zeta_b$) for selected values of $H_{max}$.**

| Reference | $H_{ref}$ criterion | $H_{ref}$* | $\zeta_b$ (10 MN) | $\zeta_b$ (15 MN) | $\zeta_b$ (20 MN) | $\zeta_b$ (25 MN) | $\zeta_b$ (30 MN) |
|---|---|---|---|---|---|---|---|
| | | [MN] | [-] | [-] | [-] | [-] | [-] |
| Peralta (2010) | $y_{ref} = 0.1 \cdot L$ | 85.1 | 0.12 | 0.18 | 0.24 | 0.29 | 0.35 |
| LeBlanc et al. (2010) | $\bar{\theta} = \theta_{ref}\sqrt{\frac{p_a}{L \cdot \gamma'}} = 4°$ | 82.1 | 0.12 | 0.18 | 0.24 | 0.30 | 0.37 |
| Klinkvort & Hededal (2013) | $\theta_{ref} = 4°$ | 65.9 | 0.15 | 0.23 | 0.30 | 0.38 | 0.46 |
| Li et al. (2015) | $y_{ref} = 0.05 \cdot D$ | 34.5 | 0.29 | 0.43 | 0.58 | 0.72 | 0.87 |
| Truong et al. (2019) | $\theta_{ref} = 0.5°$ | 19.5 | 0.51 | 0.77 | 1.03 | 1.28 | 1.54 |
| Li et al. (2020) | $y_{ref} = 0.075 \cdot D$ | 42.6 | 0.23 | 0.35 | 0.47 | 0.59 | 0.70 |

*calculated with $\gamma'$ = 10 kN/m³, $\varphi'$ = 37.5 °, $E_{s,ref}$ = 57,500 kN/m², $\lambda_{Es}$ = 0.55, $G_{o,ref}$ = 71,250 kN/m², $\lambda_{G0}$ = 0.5, $\upsilon$ = 0.225 after Thieken et al. (2015)

Table 1 shows that the majority of the listed approaches refer to the pile head displacement (y) as a deformation variable, even
if the reference load ($H_{ref}$) is partly derived from pile head rotation ($\theta$). Only LeBlanc et al. (2010) and Li et al. (2015) provide
methods for calculating the pile head rotation, whereby Li et al. (2015) propose both accumulation parameters for pile head
displacement and rotation. In order to be able to compare the individual approaches with each other, the pile head displacement
was chosen as the relevant deformation variable. To enable at least a qualitative comparison, the approach of LeBlanc et al
(2010) was therefore also applied to displacements without changing any of the model parameters, although this is not actually
permissible. Furthermore, it can be seen from Table 1 that, according to the listed approaches, only the soil relative density
($D_r$), the load magnitude ($\zeta_b$) and the cyclic load ratio ($\zeta_c$) have an influence on the model parameters. For the parameter study,
bandwidths of the mentioned parameters ($D_r$, $\zeta_b$, $\zeta_c$) were used, which lie within the application range of the examined
approaches. For the method of LeBlanc et al. (2010), the functions for $T_{b,LB}(D_r, \zeta_b)$, which in model scale apply for soil relative
densities ($D_r$) of 0.04 and 0.38, were related to relative densities ($D_r$) of 0.08 and 0.75 in true scale (cf. section 2). In cases
where there are two functions for a parameter depending on, for example, the soil relative density (see e.g. $T_{c,L}$ according to
Li et al. (2020) in Table 1) or another input value, linear interpolation was performed between the two functions as needed.
The results of the parameter study are given in Fig. 2. Here, the pile head displacements after a given number of load cycles
($y_N$) and the corresponding normalized pile head displacements ($y_N/y_1$) calculated according to the six methods summarized in
Table 1 and the before mentioned assumptions are depicted. In order to assess the influence of the different input variables
separately, only one parameter was varied and plotted on the x-axis for each diagram. When evaluating the results presented
in Fig. 2, it must be kept in mind that the approaches of Peralta (2010) and Li et al. (2015) in particular were derived for one-
way cyclic loading ($\zeta_c = 0$) only. For the sake of completeness, these are nevertheless shown in Fig. 2 (d) and (h), where the
absolute and normalized pile head displacement ($y_N$ and $y_N/y_1$) is plotted against the cyclic load ratio ($\zeta_c$).



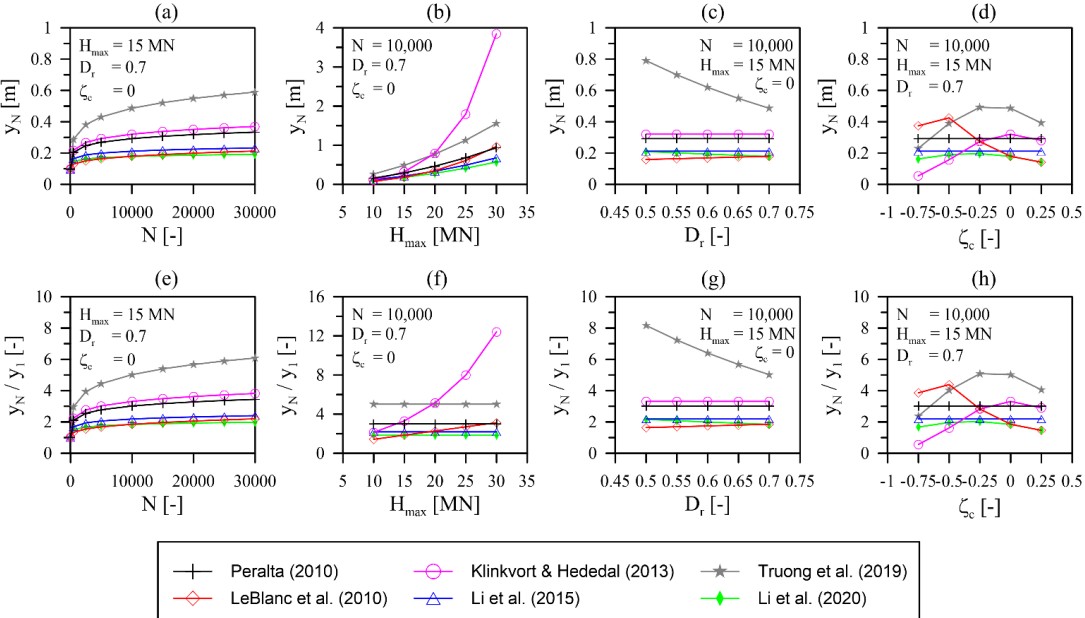

**Figure 2: Results of the parameter study on the pile head displacement accumulation and the influence of different input variables – comparison of the prediction models given in Table 1.**

For cyclic one-way loading ($\zeta_c = 0$) with $H_{max} = 15$ MN (corresponding to relative load magnitudes ($\zeta_b$) between 0.18 and 0.77, see Table 2) and a soil relative density ($D_r$) of 0.7 it can be seen from Fig. 2 (a) and (e) that according to all methods considered an accumulation of pile head displacement with increasing load cycle number (N) occurs. Lowest accumulation results from the approaches of LeBlanc et al. (2010), Li et al. (2015) and Li et al. (2020), where after 30,000 load cycles $y_N/y_1$ takes values of approximately 2.2, 2.4 and 2.0, respectively. However, it should be noted that the approach of LeBlanc et al. (2010) will result in slightly higher values ($y_N/y_1$) for further increasing load cycle numbers when compared to Li et al. (2015) and Li et al. (2020) due to its formulation (see Eq. (5)) deviating from Eq. (1) and the resulting higher accumulation rate. Largest normalized pile head displacements are predicted when applying the method of Truong et al. (2019) taking a value of $y_N/y_1 = 6.1$ and corresponding to an absolute pile head displacement ($y_N$) of 0.59 m after 30,000 load cycles. The results according to the approaches of Klinkvort and Hededal (2013) as well as Peralta (2010) fall between the results of the previously mentioned methods.

Looking at Fig. 2 (b) showing the influence of the maximum cyclic load ($H_{max}$) and thus indirectly the influence of the cyclic load magnitude ($\zeta_b$) on the pile head displacement (y) after 10.000 load cycles (N), it is obvious that in general larger absolute displacements (y) occur for higher cyclic loads ($H_{max}$). When considering the normalized pile head displacements ($y_N/y_1$) in Fig 2 (f) instead, it becomes clear that only the results according to LeBlanc et al. (2010) and Klinkvort and Hededal (2013) are affected by the maximum cyclic load ($H_{max}$) or the load magnitude ($\zeta_b$), respectively. Here, the Klinkvort and Hededal (2013) approach is much more sensitive to an increase in maximum cyclic load ($H_{max}$), even though the cyclic load magnitude ($\zeta_b$) for the chosen loads (10 MN $\leq H_{max} \leq$ 30 MN) and for the definition of the reference load ($H_{ref}$) according to this approach is in a moderate range from $0.15 \leq \zeta_b \leq 0.46$ (see Table 2). Also it should be noted that the trends and differences shown with regard to the results in the accumulation rate ($y_N/y_1$) according to Klinkvort and Hededal (2013) compared to LeBlanc et al. (2010) increase further with larger number of load cycles (N). Anyway, according to both Klinkvort and Hededal (2013) as well as LeBlanc et al. (2010) the accumulation rate ($y_N/y_1$) generally increases with increasing cyclic load magnitude ($\zeta_b$),




305 whereas this is not the case for the other approaches mentioned. Here, higher absolute displacement ($y_N$) in case of increasing maximum cyclic loads ($H_{max}$) only results from an increase in $y_1$.

Figure 2 (c) and (g) show the influence of the soil relative density ($D_r$) on the absolute and normalized pile head displacement ($y_N$ and $y_N/y_1$). From the plot, it can be seen that relative soil density ($D_r$) only has an impact on the results according to the approaches of LeBlanc et al. (2010), Truong et al. (2019) and Li et al. (2020). However, it must be kept in mind that the soil

310 relative density ($D_r$) has an influence on the soil parameters and thus on the monotonic load-displacement curve and consequently $y_1$, which was not taken into account here for reasons of comparability. Therefore, lower values would actually result for the absolute displacements ($y_N$) with increasing soil density ($D_r$). Anyway, according to both Truong et al. (2019) and Li et al: (2020), higher soil relative density ($D_r$) results in lower pile head displacement accumulation. As already shown in Fig. 2 (a) and (e), the approach according to Truong et al. (2019) here also yields the largest deformations overall. However,

315 these also decrease the most with increasing soil relative density ($D_r$), although they are still always above the other results. The results according to LeBlanc et al. (2010) are somewhat different. Here, a slight increase of the accumulated displacements with increasing soil relative density ($D_r$) can be observed. Nevertheless, the results according to LeBlanc et al. (2010) are in a similar range as those according to the other approaches with the exception of Truong et al. (2019). This is due to the fact that the soil relative density ($D_r$) according to LeBlanc et al. (2010) seems to have only minor influence.

320 In Fig. 2 (d) and (h) the influence of the cyclic load ratio ($\zeta_c$) for a maximum cyclic load ($H_{max}$) of 15 MN and a soil relative density ($D_r$) of 0.7 after 10,000 load cycles (N) is given. Irrespective of the fundamental differences in the results according to the approaches investigated, it follows from these diagrams that a variation in the cyclic load ratio ($\zeta_c$) also leads to deviating results with different overall trends. While for almost all approaches except Peralta (2010) and Li et al. (2015), who did not provide information on the influence of the cyclic load ratio ($\zeta_c$), a rather asymmetric load with $\zeta_c \leq -0.25$ results in highest

325 displacements, for the Klinkvort and Hededal (2013) approach this is true for $\zeta_c = 0$. Furthermore, it is also apparent that the different approaches are differently sensitive to the cyclic load ratio ($\zeta_c$). In particular, the results according to Li et al. (2020) stand out, in which only a marginal influence of the cyclic load ratio ($\zeta_c$) can be recognized. In contrast, the other approaches show significantly larger differences when $\zeta_c$ differs from zero.

Considering the partly deviating or even contradictory results shown above, both with respect to the absolute values ($y_N$) and

330 the trends shown with regard to the influence of the different input parameters on the accumulation rate ($y_N/y_1$), it is clear that there is a need for further research. The inconsistencies shown between the different approaches lead at best to an over dimensioning of monopile dimensions and therefore increasing costs, at worst even to uncertainties with regard to the long-time deformation behaviour of the foundation that could render the structure unsuitable for its intended function earlier than planned. One possible reason for the existing discrepancies could be, for example, the usually very limited number of tests on

335 which the various approaches are based.

## 4 Small-scale model tests

### 4.1 Objective, test program and experimental set-up

To identify and quantify the influencing parameters affecting the load displacement behaviour of a rigid pile due to lateral cyclic loading in more detail, a large campaign of small-scale 1g model tests has been designed. The parameters that have been

340 assessed include the cyclic load magnitude ($\zeta_b$), cyclic load ratio ($\zeta_c$), load eccentricity (h), soil relative density ($D_r$), the grain size distribution of the non-cohesive bedding material (soil) as well as the pile embedment length (L). In total, the entire test program, which is summarized in Table 3, comprised 15 test series (TS) with more than 150 individual tests on four different pile-soil systems in dry sand. A pile-soil system is here defined as a system with same embedment length (L), soil relative density ($D_r$), pile diameter (D) and bedding material (soil). As a model pile a tubular aluminium pipe with an outer diameter

345 (D) of 50 mm and a wall thickness (t) of 3.2 mm was used. Two different embedment lengths (L) of 300 mm (L/D = 6) and





400 mm (L/D = 8) as well as two different bedding materials (F34 and S40T, see section 4.2) with two relative densities ($D_r$) of 0.4 (medium dense) and 0.6 (dense) are considered by which the four pile-soil systems (see Table 3) are defined. According to the non-dimensional stiffness ratio suggested by Poulos and Hull (1989) all four systems can be characterized to behave almost rigid, similar to a true scale monopile. In order to investigate the influence of the load eccentricity (h) or the ratio of horizontal force to overturning moment respectively, the ratio of load eccentricity (h) to embedment length (L) was varied in the range of h/L = 0.6 to h/L = 1.0 for pile-soil system 1 and h/L = 0.8 to h/L = 1.2 for pile-soil system 4. For pile-soil systems 2 and 3 the ratio (h/L) was kept constant taking a value of 1.0.

**Table 3: Test program.**

| | | Pile-soil system description | | | | | | Load description | |
|---|---|---|---|---|---|---|---|---|---|
| Test series | System | D | L/D | h/L | Soil | $D_r$ | $\zeta_b$ | $\zeta_c$ | N |
| [#] | [#] | [mm] | [1] | [1] | [-] | [1] | [1] | [1] | [-] |
| 1 | 1 | 50 | 8 | 0.6 | F34 | 0.4 | 0.35 | -0.75/-0.50/-0.25/0.00/0.25/1.00 | 2500 |
| 2 | 1 | 50 | 8 | 0.8 | F34 | 0.4 | 0.35 | -0.75/-0.50/-0.25/0.00/0.25/1.00 | 2500 |
| 3 | 1 | 50 | 8 | 1.0 | F34 | 0.4 | 0.15 | -0.75/-0.50/-0.25/0.00/0.25/1.00 | 2500 |
| 4 | 1 | 50 | 8 | 1.0 | F34 | 0.4 | 0.25 | -0.75/-0.50/-0.25/0.00/0.25/1.00 | 2500 |
| 5 | 1 | 50 | 8 | 1.0 | F34 | 0.4 | 0.35 | -0.75/-0.50/-0.25/0.00/0.25/1.00 | 2500 |
| 6 | 2 | 50 | 8 | 1.0 | F34 | 0.6 | 0.15 | -0.75/-0.50/-0.25/0.00/0.25/1.00 | 2500 |
| 7 | 2 | 50 | 8 | 1.0 | F34 | 0.6 | 0.25 | -0.75/-0.50/-0.25/0.00/0.25/1.00 | 2500 |
| 8 | 2 | 50 | 8 | 1.0 | F34 | 0.6 | 0.35 | -0.75/-0.50/-0.25/0.00/0.25/1.00 | 2500 |
| 9 | 3 | 50 | 8 | 1.0 | S40T | 0.4 | 0.15 | -0.75/-0.50/-0.25/0.00/0.25/1.00 | 2500 |
| 10 | 3 | 50 | 8 | 1.0 | S40T | 0.4 | 0.25 | -0.75/-0.50/-0.25/0.00/0.25/1.00 | 2500 |
| 11 | 3 | 50 | 8 | 1.0 | S40T | 0.4 | 0.35 | -0.75/-0.50/-0.25/0.00/0.25/1.00 | 2500 |
| 12 | 4 | 50 | 6 | 0.8 | F34 | 0.4 | 0.20 | -0.75/-0.50/-0.25/0.00/0.25/1.00 | 2500 |
| 13 | 4 | 50 | 6 | 0.8 | F34 | 0.4 | 0.35 | -0.75/-0.50/-0.25/0.00/0.25/1.00 | 2500 |
| 14 | 4 | 50 | 6 | 1.0 | F34 | 0.4 | 0.35 | -0.75/-0.50/-0.25/0.00/0.25/1.00 | 2500 |
| 15 | 4 | 50 | 6 | 1.2 | F34 | 0.4 | 0.35 | -0.75/-0.50/-0.25/0.00/0.25/1.00 | 2500 |

The loading conditions in the model tests comprised both displacement controlled monotonic loading tests ($\zeta_c = 1$) and load controlled cyclic loading (sinusoidal) with different cyclic load magnitudes ($\zeta_b$) and cyclic load ratios ($\zeta_c$) at a constant frequency of 0.1 Hz with 2500 load cycles (N) each. The cyclic load magnitude ($\zeta_b$) was defined according to Eq. (3) based on a reference load ($H_{ref}$) that has been defined for each configuration of pile-soil system and load eccentricity (h) from the determined monotonic load-displacement curves by application of a pile failure criterion (see section 4.3.1). To evaluate the influence of different cyclic load ratios ($\zeta_c$) as given in Eq. (4), one- and two-way loads with values of $\zeta_c = -0.75/-0.50/-0.25/0.00/+0.25$ have been applied. In order to increase the significance of the experimental results and to confirm repeatability, all tests given in Table 3 have at least been conducted twice.

The small-scale model tests were carried out on a specially designed test rig, consisting of a sand container, a model pile, an electromechanical actuator and several sensors. Figure 3 shows the schematic structure of the experimental set-up, its dimensions and its individual components. For more detailed information on the test equipment or scaling considerations it is referred to Frick and Achmus (2020). Physical quantities measured in y-direction (see Fig. 3), such as displacements or forces imposed by pulling with the actuator, are positive in the following.



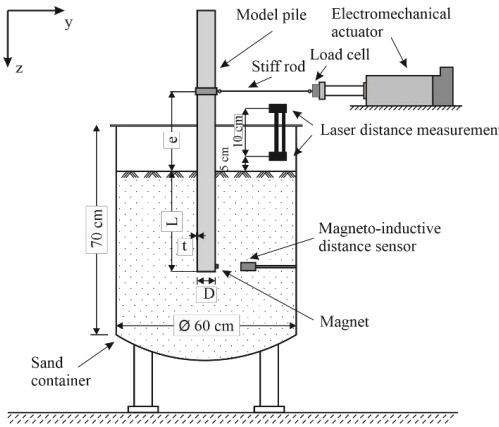


**Figure 3: Schematic of experimental test set-up.**

### 4.2 Soil properties and sand sample preparation

For the model test campaign, two different non-cohesive bedding materials where chosen to investigate the influence of the grain size distribution to the pile response due to lateral cyclic loading. Both materials are commercially available silica sands with the designations F34 and S40T. The F34 is a fine to medium grained sand having a mean effective particle size ($d_{50}$) of

0.18 mm and a uniformity coefficient ($C_U$) of 1.90. In contrast, the S40T is a coarse sand ($d_{50} = 0.82$ mm) with a slightly narrower graded grain size distribution ($C_U = 1.4$). For both bedding materials the sand characteristics and peak friction angles ($\varphi'_{peak}$) determined from standard shear box tests with normal stresses ($\sigma_v$) of 20, 40 and 80 kN/m² and soil relative densities ($D_r$) of 0.4 are given in Table 4. The grain size distributions of both soils are depicted in Fig. 4.

**Table 4: Properties of F34 and S40T silica sands.**

| Description | Parameter | Unit | Value |
|---|---|---|---|
| | | | F34 / S40T |
| Mean grain size | $d_{50}$ | [mm] | 0.18 / 0.82 |
| Uniformity coefficient | $C_U$ | [1] | 1.90 / 1.40 |
| Coefficient of curvature | $C_C$ | [1] | 1.02 / 1.00 |
| Minimum void ratio | $e_{min}$ | [1] | 0.585 / 0.481 |
| Maximum void ratio | $e_{max}$ | [1] | 0.887 / 0.751 |
| Grain density | $\rho_s$ | [g/cm³] | 2.65 / 2.65 |
| Peak angle of friction | $\varphi'_{peak}(D_r=0.4)$ | [°] | 34.7 / 32.2 |


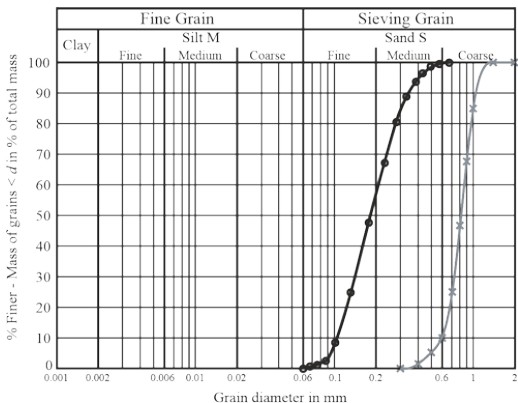

**Figure 4: Grain size distribution of F34 (black) and S40T (grey) silica sands.**





Before each small-scale model test a uniform sand sample with one of the two chosen soil relative densities ($D_r$) had to be prepared. This was done by air pluviation through a series of diffusor meshes and with a defined drop height. In a series of
preliminary tests, appropriate meshes and drop heights were determined, with the help of which the desired soil relative densities ($D_r$) for the two different bedding materials (see Table 3) could be achieved. Furthermore, it could be shown that the selected preparation procedure leads to homogeneous and reproducible sand conditions in a normally consolidated state. To avoid complex stress fields and local changes in soil density due to model pile installation to influence the test results, it was decided to omit the pile installation procedure from the model tests and prepare the sand around the pre-installed pile.
Nevertheless, in order to allow mobilization of shear stresses at the pile toe, the sand container was first filled to a height of about 5 cm above the later position of the pile toe before the pile was placed in the container, slightly pushed into the 5 cm thick sand bed and fixed in position by a clamping system. The remaining soil preparation was then carried out around the already installed pile. The soil dry unit weight ($\gamma$) resulting from this preparation procedure for the S40T sand with a relative density ($D_r$) of 0.4 is 16.1 kN/m³. For the F34 sand, the soil dry unit weight ($\gamma$) is 15.0 kN/m³ and 15.5 kN/m³ for relative
densities ($D_r$) of 0.4 and 0.6, respectively.

### 4.3 Test results

#### 4.3.1 Monotonic test results

In order to be able to apply comparable load conditions in terms of the load magnitude ($\zeta_b$) in the cyclic tests despite different pile-soil systems and lever arms (h), first monotonic load tests were carried out for the determination of load-displacement
curves and a reference load ($H_{ref}$) for each configuration. Fig. 5 presents the variations of normalised monotonic lateral load ($H/(\gamma \cdot L^3)$) with normalized pile head displacement (y/L) at soil surface for all four pile-soil systems and in case of pile-soil system 1 and 4 for different load eccentricities (h) additionally. Here, the pile head displacements (y) calculated from the measured displacements of the two laser distance transducers (see Fig. 3) are depicted as solid lines. To ensure the reproducibility of the tests and to check the quality of the sand sample preparation, each test was conducted at least twice. As
some scattering could be observed especially for pile-soil system 2 (h/L = 1) and pile-soil system 4 with a normalized load eccentricity (h/L) of 1.2, these tests have been done even four times.

For this study, the reference load ($H_{ref}$) should be defined as the ultimate lateral pile capacity ($H_{ult}$) at total pile failure. The failure load ($H_{ult}$) that can be resisted by a rigid pile is a function of the ultimate lateral resistance that can be mobilized by the soil against the pile. The mobilized soil resistance in case of a laterally loaded rigid pile is again characterized by two failure
mechanisms. The first occurs at shallow depths and is due to the formation of a passive wedge in front of the pile and in the direction of loading. The second is associated with the plastic flow of soil around the pile in the horizontal plane at greater depths. For the occurrence of both failure mechanisms and thus the mobilization of the full soil resistance against the pile, very large displacements are required. From the monotonic test results depicted in Fig. 5 it emerges that total pile capacity ($H_{ult}$) defined by full soil plastification and a load-displacement curve approaching a horizontal tangent has not been reached despite
very large displacements. In order to be able to determine the pile capacities ($H_{ult}$) and therefore the chosen reference loads ($H_{ref}$) of the individual pile-soil systems, the method of Manoliu et al. (1985) was applied to the results. This method assumes that load-displacement of a laterally loaded pile can be described by a hyperbolic function. Therefore, the method allows estimation of the pile failure load ($H_{ult}$) by extrapolation of measured test data. Corresponding extrapolation curves derived by application of this approach are depicted in Fig. 5 as dashed lines.



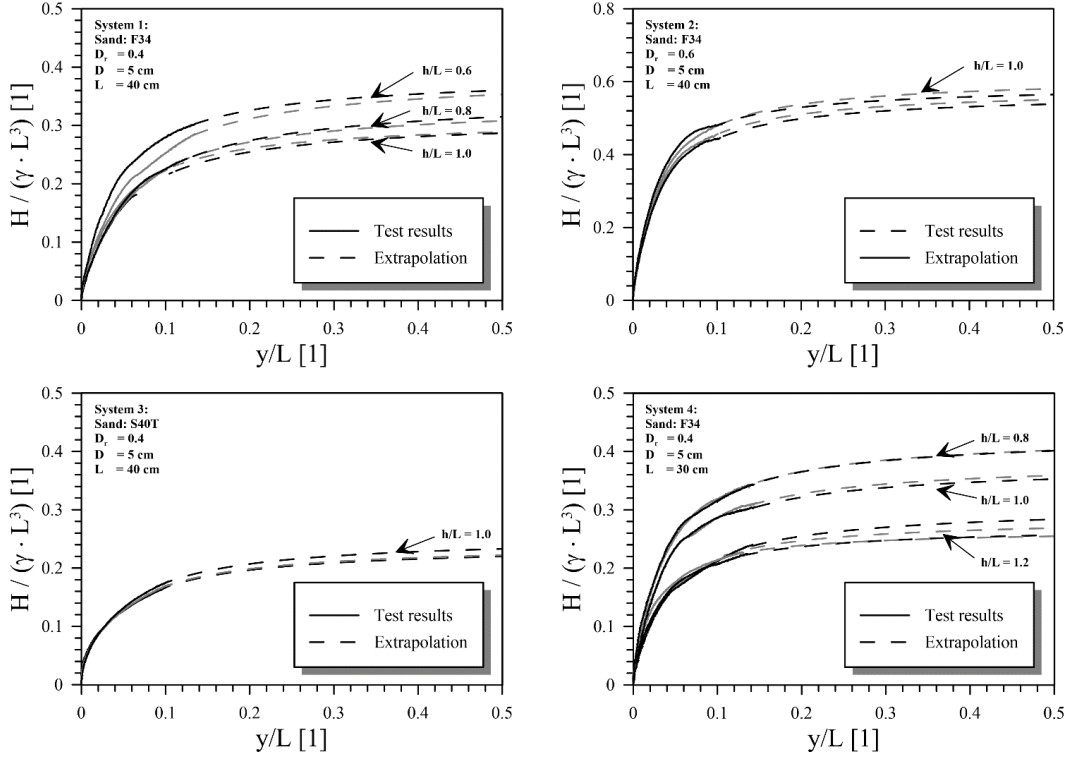


**Figure 5: Load-displacement curves from monotonic lateral load tests and extrapolations according to Manoliu et al. (1985).**

As some slight scattering could be observed in the monotonic test results and extrapolations (Fig. 5), mean values for the pile failure load ($H_{ult}$) or reference load ($H_{ref}$), respectively, were used for each configuration to define the cyclic load magnitude ($\zeta_b$) according to Eq. 3 for each configuration. The normalized and absolute reference values of the horizontal load ($H_{ref}$)

determined for each test configuration according to the previously mentioned procedure are summarized in Table 5.

**Table 5: Pile reference loads ($H_{ref}$) determined by extrapolation of test results.**

| System | h/L | $H_{ref}/(\gamma \cdot L^3)$ | $H_{ref}$ |
|---|---|---|---|
| [#] | [1] | [1] | [N] |
| 1 | 0.6 | 0.388 | 372.0 |
| | 0.8 | 0.344 | 330.0 |
| | 1.0 | 0.312 | 299.7 |
| 2 | 1.0 | 0.589 | 583.8 |
| 3 | 1.0 | 0.245 | 252.0 |
| 4 | 0.8 | 0.430 | 174.0 |
| | 1.0 | 0.380 | 154.0 |
| | 1.2 | 0.282 | 114.4 |

**4.3.2 Cyclic test results**

The cyclic tests summarized in Table 3 have been conducted for cyclic load magnitudes ($\zeta_b$) of 0.15, 0.25 as well as 0.35 and cyclic load ratios ($\zeta_c$) ranging from nearly balanced two-way loading ($\zeta_c = -0.75$) to one-way loading with complete unloading

($\zeta_c = 0.00$) or partial unloading ($\zeta_c = +0.25$) in each cycle. Based on the findings of Jalbi et al. (2019), who proposed a practical method to predict the nature of monopile loading conditions ($\zeta_b$ and $\zeta_c$) and evaluated 15 existing wind turbines in Europe using their method, the load magnitudes ($\zeta_b$) and cyclic load ratios ($\zeta_c$) selected for this study are of particular interest for practical application. According to Jalbi et al. (2019) typical load magnitudes ($\zeta_b$) for normal operational conditions range from



0.1 to 0.2. In extreme wind and wave loading cases also load magnitudes ($\zeta_b$) of up to 0.4 may be reached. With regard to the cyclic load ratio ($\zeta_c$) they found that loads on monopiles are mostly one-way ($\zeta_c \geq 0$) under normal operational conditions but may also be two-way ($\zeta_c < 0$) in extreme loading scenarios, especially in deep waters. It should be mentioned that Jalbi et al. (2019) also assumed the reference load ($H_{ref}$) for the definition of the load magnitude ($\zeta_b$) by back-calculations, where ultimate pressure of the ground profile was mobilized (= total ground and therefore pile failure), so that values given for $\zeta_b$ should be comparable with those of this study (see section 4.3.1).

In Fig. 6, the results of all cyclic tests are plotted in terms of normalized pile head displacement ($y_N/y_1$) against load cycle number (N) for each of the 15 test series separately. In addition, power functions according to Eq. 1 have been fitted to the measured test results and are also shown in Fig. 6. The underlying maximum pile head displacement after application of the first load cycle ($y_1$) as well as the determined accumulation parameter ($\alpha$) in dependence on the cyclic load ratio ($\zeta_c$) for each individual test, is listed in Table 6 for clarity.

In general, it can be seen from Fig. 6 that the tests are well reproducible in most cases. Furthermore, it can be seen that the measured curves can be described very well by the selected power function (Eq. 1). Only in a few cases (see, e.g. test series 5: $\zeta_c$ = -0.25) there seems to be a minimal overestimation of measured values as the number of load cycles increases. With respect to the influence of the cyclic load ratio ($\zeta_c$), a clear trend can be recognized in all test series. Irrespective of the load eccentricity (h) or the pile embedment length (L), for pile soil systems 1 (test series 1-5), 3 (test series 9-11) and 4 (test series 12-15), the

highest displacement accumulation consistently results from an unbalanced two- way loading with a cyclic load ratio ($\zeta_c$) of -0.25. In case of pile-soil system 3 (test series 9-11) with the S40T sand as bedding material and a soil relative density ($D_r$) of 0.4, however, the difference between the tests with $\zeta_c$ = -0.25 and $\zeta_c$ = 0.00 is less pronounced. Also for pile-soil system 2 (test series 6-8) with dense ($D_r$ = 0.6) F34 sand the largest displacements result for an asymmetric two-way loading, however not with a cyclic load ratio ($\zeta_c$) of -0.25, but at -0.5. The lowest displacement accumulations for all test series result from loading

with a cyclic load ratio ($\zeta_c$) of -0.75 (nearly balanced two-way loading) or +0.25 (one-way loading without complete unloading in each cycle). A negative accumulation for loads with large negative cyclic load ratios ($\zeta_c$) as reported by Klinkvort and Hededal (2013) could not be observed, although very small accumulations were recorded in some cases for the tests with $\zeta_c$ = -0.75 (see, e.g. test series 12 &13). Another general trend that emerges from the results shown in Fig. 6 is that a large part of the total deformations due to lateral cyclic loading already takes place within the first 500 to 1000 load cycles, while

subsequently there is a slowly decreasing accumulation rate (sedation). An exception here is test series 8 (F34, $D_r$ = 0.6 and $\zeta_b$ = 0.35), where the test results with negative cyclic load ratios ($\zeta_c$) show a strong increase in displacements even beyond a cycle number (N) of 1000. This is due to the fact that in this configuration with high cyclic loads (large $H_{ref}$ and $\zeta_b$), especially with alternating loads ($\zeta_c < 0$), the pile moved slowly out of the soil while cycling, resulting in progressive failure. This is also the reason why no results are shown for cyclic load ratios ($\zeta_c$) of -0.75 for this test series, as here an even earlier failure

occurred. Having this is mind, the results for the pile-soil system 2 (test series 6-8) and especially those of test series 8 should be treated with caution. For the above reasons, the results of test series 8 have been shaded grey in Table 6.







**Figure 6: Cyclic test results – Normalized pile head displacement ($y_N/y_1$) against load cycle number (N) for all 15 test series.**





**Table 6: Cyclic test results - Determined accumulation parameters α(ζ_c) and measured pile head displacements after first loading (y₁).**

| Test series | D | L/D | h/L | Soil | D_r | ζ_b | α(ζ_c) [1] / $y_1$ [mm] | | | | | | | | | |
|---|---|---|---|---|---|---|---|---|---|---|---|---|---|---|---|---|
| [#] | [mm] | [1] | [1] | [-] | [1] | [1] | -0.75 | -0.75 | -0.50 | -0.50 | -0.25 | -0.25 | 0.00 | 0.00 | +0.25 | +0.25 |
| 1 | 50 | 8 | 0.6 | F34 | 0.4 | 0.35 | 0.059 | 0.068 | 0.124 | 0.118 | 0.146 | 0.138 | 0.118 | 0.123 | 0.093 | 0.085 |
|   |   |   |   |   |   |   | 9.90 | 8.94 | 8.20 | 9.18 | 9.74 | 9.48 | 8.72 | 8.50 | 7.15 | 8.30 |
| 2 | 50 | 8 | 0.8 | F34 | 0.4 | 0.35 | 0.065 | 0.059 | 0.097 | 0.107 | 0.135 | 0.140 | 0.134 | 0.128 | 0.087 | 0.079 |
|   |   |   |   |   |   |   | 11.92 | 9.36 | 11.85 | 10.52 | 10.49 | 8.00 | 7.51 | 8.98 | 9.87 | 10.01 |
| 3 | 50 | 8 | 1.0 | F34 | 0.4 | 0.15 | - | 0.065 | 0.123 | 0.133 | 0.149 | 0.142 | 0.108 | 0.106 | 0.075 | 0.079 |
|   |   |   |   |   |   |   | - | 1.93 | 2.21 | 2.02 | 2.06 | 2.45 | 2.19 | 2.34 | 3.21 | 2.79 |
| 4 | 50 | 8 | 1.0 | F34 | 0.4 | 0.25 | 0.102 | 0.107 | 0.130 | 0.125 | 0.153 | 0.142 | 0.117 | 0.112 | 0.079 | 0.084 |
|   |   |   |   |   |   |   | 4.35 | 4.51 | 4.96 | 4.66 | 4.08 | 4.24 | 4.40 | 4.55 | 4.54 | 3.99 |
| 5 | 50 | 8 | 1.0 | F34 | 0.4 | 0.35 | 0.107 | 0.079 | 0.129 | 0.117 | 0.158 | 0.151 | 0.120 | 0.131 | 0.082 | 0.089 |
|   |   |   |   |   |   |   | 6.86 | 7.38 | 6.95 | 7.60 | 6.70 | 7.44 | 9.40 | 7.30 | 7.61 | 6.85 |
| 6 | 50 | 8 | 1.0 | F34 | 0.6 | 0.15 | 0.104 | 0.129 | 0.161 | 0.135 | 0.149 | 1.143 | 0.122 | 0.098 | 0.078 | 0.068 |
|   |   |   |   |   |   |   | 1.65 | 2.18 | 1.52 | 1.87 | 1.73 | 1.77 | 1.65 | 1.89 | 1.60 | 1.91 |
| 7 | 50 | 8 | 1.0 | F34 | 0.6 | 0.25 | 0.119 | 0.115 | 0.176 | 0.174 | 0.164 | 0.157 | 0.125 | 0.116 | 0.066 | 0.091 |
|   |   |   |   |   |   |   | 3.23 | 4.07 | 3.80 | 4.18 | 3.58 | 3.83 | 3.94 | 3.80 | 3.86 | 3.78 |
| 8 | 50 | 8 | 1.0 | F34 | 0.6 | 0.35 | - | - | 0.206 | 0.219 | 0.191 | 0.185 | 0.131 | 0.133 | 0.103 | - |
|   |   |   |   |   |   |   | - | - | 5.73 | 4.81 | 5.62 | 5.22 | 5.84 | 5.74 | 5.16 | - |
| 9 | 50 | 8 | 1.0 | S40T | 0.4 | 0.15 | 0.099 | 0.073 | 0.108 | 0.119 | 0.119 | 0.142 | 0.111 | 0.122 | 0.091 | 0.082 |
|   |   |   |   |   |   |   | 1.62 | 1.75 | 1.90 | 1.55 | 1.85 | 1.57 | 1.89 | 1.49 | 1.39 | 1.46 |
| 10 | 50 | 8 | 1.0 | S40T | 0.4 | 0.25 | 0.040 | 0.046 | 0.098 | 0.101 | 0.115 | 0.112 | 0.116 | 0.108 | 0.077 | 0.077 |
|   |   |   |   |   |   |   | 4.45 | 4.59 | 5.14 | 5.13 | 5.52 | 5.12 | 5.27 | 5.41 | 4.95 | 5.42 |
| 11 | 50 | 8 | 1.0 | S40T | 0.4 | 0.35 | 0.041 | 0.041 | 0.084 | 0.086 | 0.103 | 0.101 | 0.101 | 0.108 | 0.081 | 0.083 |
|   |   |   |   |   |   |   | 11.50 | 10.73 | 11.52 | 10.85 | 11.25 | 11.25 | 10.54 | 10.21 | 10.36 | 10.31 |
| 12 | 50 | 6 | 0.8 | F34 | 0.4 | 0.20 | 0.029 | 0.009 | 0.098 | 0.100 | 0.148 | 1.142 | 0.119 | 0.129 | 0.085 | 0.076 |
|   |   |   |   |   |   |   | 2.16 | 2.61 | 3.26 | 2.83 | 2.99 | 2.63 | 2.73 | 2.52 | 2.56 | 3.43 |
| 13 | 50 | 6 | 0.8 | F34 | 0.4 | 0.35 | 0.020 | 0.008 | 0.106 | 0.099 | 0.133 | 0.132 | 0.109 | 0.111 | 0.075 | 0.081 |
|   |   |   |   |   |   |   | 5.65 | 6.27 | 6.59 | 8.40 | 7.05 | 6.45 | 8.23 | 6.72 | 6.65 | 6.92 |
| 14 | 50 | 6 | 1.0 | F34 | 0.4 | 0.35 | 0.062 | 0.067 | 0.086 | 0.093 | 0.129 | 0.149 | 0.117 | 0.119 | 0.085 | 0.076 |
|   |   |   |   |   |   |   | 6.72 | 5.90 | 5.72 | 5.41 | 6.39 | 6.18 | 5.81 | 5.62 | 5.26 | 5.86 |
| 15 | 50 | 6 | 1.2 | F34 | 0.4 | 0.35 | 0.060 | 0.054 | 0.096 | 0.098 | 0.121 | 0.125 | 0.109 | 0.107 | 0.080 | 0.078 |
|   |   |   |   |   |   |   | 4.62 | 5.71 | 4.32 | 5.09 | 5.62 | 5.09 | 5.36 | 6.28 | 5.10 | 4.91 |

### 4.3.3 Evaluation

In order to evaluate the cyclic test results from Fig. 6 with respect to the influence of the applied cyclic loading conditions or different parameters of the pile-soil system on the displacement accumulation, the accumulation parameters (α) for cyclic one-way loading (ζ_c = 0) from Table 6 were used as a reference value and plotted against the variable parameters of the investigated

 pile-soil systems (D_r, h/L, L/D) and the load magnitude (ζ_b) in Fig. 7. In general it is evident from Fig. 7 that the determined accumulation parameters for cyclic one-way loading (α(ζ_c = 0)) are subjected to a certain degree of unsystematic scattering, ranging from a maximum value of 0.1343 to a minimum value of 0.0983. The mean of all α-values for cyclic one-way loading (ζ_c = 0) is 0.1169 (see Table 6). The deviations in the results for the individual tests with identical boundary conditions (redundant tests) are generally smaller, but they are nevertheless present and probably due to experimental scatter.

 In Fig. 7 (a) the influence of the cyclic load magnitude (ζ_b) can be seen. When taking into account only the accumulation parameters α(ζ_c = 0) for pile-soil system 1 (black symbols) or 2 (green symbols) a slight increase of the accumulation parameter (α) with cyclic load magnitude (ζ_b) can be observed. Anyway, the opposite is true for systems 3 (blue symbols) and 4 (red symbols), respectively. When all pile-soil systems are considered in a linear regression analysis, the aforementioned trends almost cancel each other out, resulting in only negligible increase of the accumulation parameter (α) with cyclic load magnitude

 (ζ_b). Anyway, the linear equation describing the possibly existing dependency of the accumulation parameter (α) on the cyclic load magnitude (ζ_b) is given in Fig. 7 (a) for completeness.

An evaluation of the results with respect to the soil relative density (D_r) as shown in Fig. 7 (b) also does not allow any clear conclusion to be drawn. On the one hand, only one system with a higher soil relative density (D_r) was investigated and, on the other hand, the results of the individual pile-soil systems scatter over such a range that the linear regression shown in Fig. 7

 (b) can only provide an approximation. Similar to the influence of the cyclic load magnitude (ζ_b), a trend of a slightly increasing



accumulation parameter (α) with soil relative density can be seen, but this increase is also considered negligible due to the existing scatter and the minor slope of the regression curve.

In Fig. 7 (c) all accumulation parameters for cyclic one-way loading ($\alpha(\zeta_c = 0)$) are plotted against the relative load eccentricity (h/L). Taking all values into account a decreasing trend with increasing load eccentricity (h) results from the linear regression

495 analysis. If only the values for system 1 (black symbols) or system 3 (blue symbols) are considered, for which the load eccentricity was varied, it becomes clear that this behaviour is also more likely to be due to experimental scattering. For system 1, where the normalized lever arm (h/L) was varied in the range of 0.6 to 1.0, the largest accumulation parameter (α) was determined for the mean value of h/L = 1.0. For system 3 ($0.8 \leq h/L \leq 1.2$) a similar observation can be made. Since the lever arm (h) only defines the ratio of the horizontal force to the applied overturning moment, a maximum value of the accumulation

500 parameter (α) in the middle of the investigated bandwidth for the normalized load eccentricity (h/L) is not plausible.

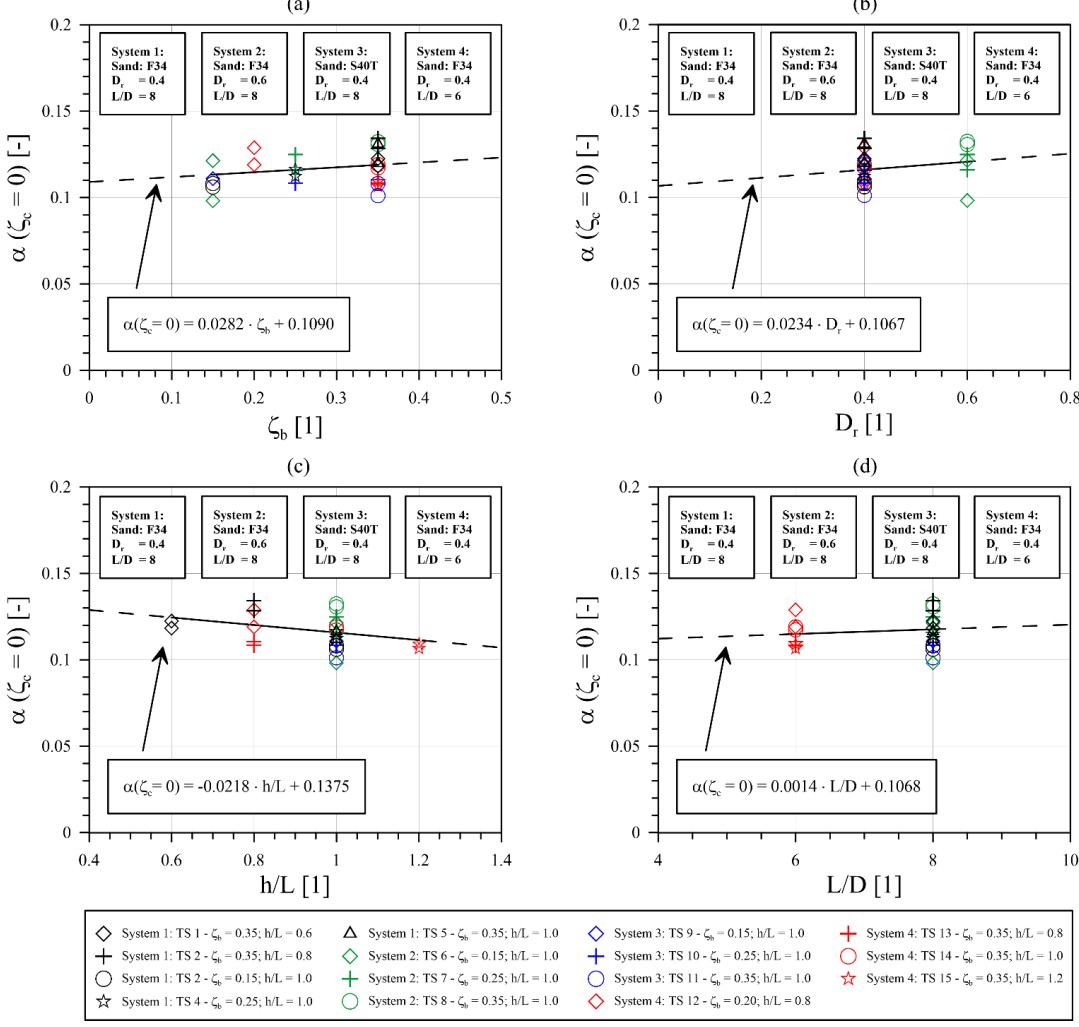

**Figure 7: Evaluation of cyclic test results – Relationship of the accumulation parameter α for $\zeta_c = 0$ with (a) load magnitude $\zeta_b$, (b) soil relative density $D_r$, (c) normalized load eccentricity h/L and (d) normalized embedment length L/D.**

Finally, Fig. 7 (d) shows the determined accumulation parameters (α) for $\zeta_c = 0$ as a function of the normalized pile embedment

505 length (L/D). As for the other investigated parameters ($\zeta_b$, $D_r$, h/L), there is no appreciable influence of the pile embedment



length (L) on the accumulation parameter (α) at least for the rigid piles in the investigated range of normalized embedment length (L/D) and for one-way cyclic loading ($\zeta_c = 0$).

Although no clear trend emerges from any of the graphs on Fig. 7 in view of the scatter present, a linear regression analysis was performed for each plot. The resulting equations describing the determined and aforementioned dependencies are given in the respective diagrams for completeness. Due to the insignificance of the observed dependencies combined with the existing variance of the results, it seems that there is no remarkable influence of the investigated parameters on the accumulation parameter (α) at least for one-way cyclic loading ($\zeta_c = 0$). Nevertheless, it should be kept in mind that at least the initial displacement ($y_1$) depends strongly on the mentioned load or pile-soil system parameters, which is why the absolute accumulated displacement after a certain number of load cycles ($y_N$) is of course not independent of the mentioned input variables (see Eq.1).

Further, Fig. 8 shows the accumulation parameter (α) and its dependency on the cyclic load ratio ($\zeta_c$) for all four pile-soil systems. Here, the results for test series 8 with negative cyclic load ratios ($\zeta_c < 0$) have been excluded from evaluation due to before mentioned reason (see section 4.3.2). Furthermore, due to the previously determined predominantly independence of the accumulation parameter for cyclic one-way loading ($\alpha(\zeta_c = 0)$), it was decided not to normalize the accumulation parameters ($\alpha(\zeta_c)$) on the basis of one of the other parameters (e.g. $\zeta_b$ or $D_r$), as suggested for example by Klinkvort and Hededal (2013), Truong et al. (2019) or Li et al. (2020). Instead, the results in Fig. 8 are enveloped by two functions defining an upper and lower bound of the accumulation parameter (α) for the investigated pile-soil systems and boundary conditions, illustrating the possible range of α-values.

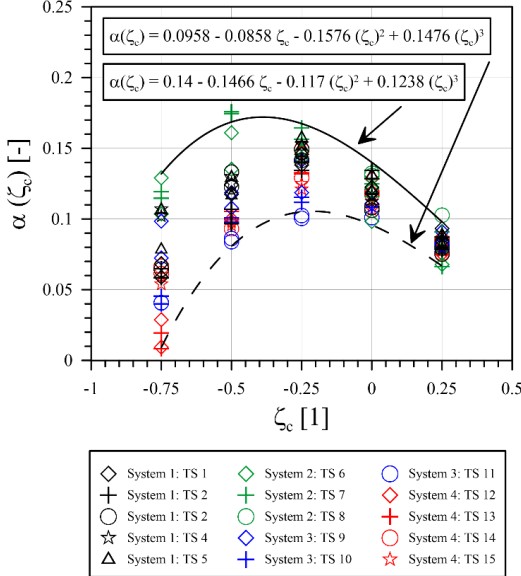

**Figure 8: Evaluation of cyclic test results – Relationship between accumulation parameter (α) with cyclic load ratio ($\zeta_c$) and corresponding lower and upper bound curves.**

In general, it can be seen from Fig. 8 that largest values for the accumulation parameter (α) result from unbalanced two-way loading ($\zeta_c < 0$) taking a maximum value of approximately 0.17 for pile-soil system 2 at a cyclic load ratio ($\zeta_c$) of -0.5 and being more or less independent from cyclic load magnitude ($\zeta_b$). On closer examination, it emerges that for all other pile-soil systems (1, 3 and 4) the maximum accumulation parameter (α) occurs with a lower value at a cyclic load ratio ($\zeta_c$) of -0.25. It could be concluded that both the maximum of the accumulation parameter (α) and its occurrence with respect to the cyclic load ratio ($\zeta_c$) depend on the soil relative density ($D_r$). In the investigated cases, an increase of the soil relative density ($D_r$) from 0.4 (System 1) to 0.6 (System 2) leads to a slight increase and simultaneous shift of the maximum accumulation parameter (α)



towards a more negative value of the cyclic load ratio ($\zeta_c$). Anyway, as already described in section 4.3.2, the results for pile-
535 soil system 2 should be handled with care. When considering only the results for pile-soil systems 1, 3 and 4, a certain spread
of the determined accumulation parameters ($\alpha$) is still evident, but basically they follow a consistent trend. Within the above-
mentioned range of values for pile-soil systems 1, 3 and 4, the values for system 1 in particular are at the upper bound, while
the accumulation parameters ($\alpha$) for systems 3 and 4 tend to be below this. Especially for a cyclic load ratio ($\zeta_c$) of -0.75 the
accumulation parameters ($\alpha$) for system 4 with a shorter embedment length partially lie in a very low range. Due to the
540 scattering of the results, a clear final conclusion cannot be drawn. However, it is evident from the results that both the
embedment length (L) of the pile (compare System 1 and System 4 in Fig. 8) and the grain size distribution (compare System
1 and System 3 in Fig. 8) appear to have an effect on the accumulation parameter ($\alpha$).

## 5 Discussion

In this section the findings and results from the conducted experimental 1g model test campaign are discussed and compared
with those of other research groups so that a classification of the results is possible. With respect to the accumulation parameter
($\alpha$) from Eq. 1, the results indicate that it appears to be largely independent of the cyclic load magnitude ($\zeta_b$), the soil relative
density ($D_r$), the load eccentricity (h) and the embedment length of the pile (L) for one-way cyclic loading ($\zeta_c = 0$), as long as
the pile-soil system is characterised by an almost rigid load-displacement behaviour. Despite some scattering in the results for
the accumulation parameter ($\alpha$), which could probably be due to irregularities in the test execution (soil sample preparation,
etc.), a mean value of $\alpha_{mean}(\zeta_c = 0) = 0.1169$ (with $\alpha_{min}(\zeta_c = 0) = 0.0983$ and $\alpha_{max}(\zeta_c = 0) = 0.1343$) could be determined. This
mean value fits quite good the value of $\alpha_P = 0.12$ proposed by Peralta (2010) who also determined it from scaled 1g model
tests on rigid piles subjected to cyclic one-way loading ($\zeta_c = 0$) only. Similar to the present study, Peralta (2010) also found
the accumulation parameter ($\alpha(\zeta_c = 0)$) to be independent from cyclic load magnitude ($\zeta_b$), the soil relative density ($D_r$) and the
normalized pile embedment length (L/D) as long as the pile behaves almost rigid.
In contrast, Li et al. (2015) proposed a lower value of $\alpha_{y,L} = 0.085$ (see Table 1) for cyclic one-way loading ($\zeta_c = 0$) based on
two cyclic laterally loaded field tests on rigid piles. This could indicate that accumulation parameters ($\alpha$) from small-scale
model tests cannot be easily transferred to true scale due to differences in the stress state of the surrounding soil and the
resulting differences in soil behaviour (e.g. dilatancy, stiffness, etc.). This assumption can be supported by Richards et al.
(2021) who investigated the stress effect on the response of model monopiles to unidirectional cyclic lateral loading ($\zeta_c = 0$)
in sand by model tests either at 1g or in the centrifuge. Although in this study an approximation function according to Eq. 5
was used to describe the cyclic displacement behaviour, it was found that the accumulation parameter ($\alpha$) of this equation (not
directly comparable with $\alpha$ according to Eq. 1) decreases logarithmically with stress level under otherwise constant boundary
conditions. Qualitatively, according to Richards et al. (2021) the cyclic responses have been found to be similar across stress-
level, anyway. It is therefore obvious that the present results provide higher accumulation parameters ($\alpha$) than comparable
large-scale experiments or centrifuge tests at higher stress-levels.
This can also be verified using the approach of Li et al. (2020), which is based on a series of centrifuge tests. According to this
approach, the accumulation parameter ($\alpha$) from Eq. 1 can be calculated by multiplication of the two parameters $T_{b,L}$ and $T_{c,L}$
that describe the influence of the load magnitude ($\zeta_b$) as well as the cyclic load ratio ($\zeta_c$) together with the soil relative density
($D_r$) respectively (for corresponding equations see Table 1). Due to the definition of $T_{c,L}$ according to Eq. 9 together with Eq 8,
for cyclic one-way loading ($\zeta_c = 0$) the accumulation parameter ($\alpha$) for this approach would have to result in $T_{b,L} = 0.07335$
when the proposed functions for $T_{b,L}$ and $T_{c,L}$ would fit the underlying test results perfectly.

$$T_{b,L}(\zeta_b) = \alpha(\zeta_c = 0, \zeta_b) = 0.07335 \tag{8}$$

$$T_{c,L}(\zeta_c) = \frac{\alpha(\zeta_c)}{T_{b,L}(\zeta_b)} \tag{9}$$





Since the results of Li et al. (2020), on which the approach and the functions are based, are also subjected to scattering, the
constant value of $T_{b,L}$ is only an approximation of the experimentally determined accumulation parameters ($\alpha(\zeta_c = 0)$), which
is why the proposed $T_{c,L}$-functions (see Table 1) do not yield the value of 1 when the cyclic load ratio ($\zeta_c$) is 0. Nevertheless,
the approach yields accumulation parameters ($\alpha$) of about (+/-) 0.07335 for cyclic one-way loading ($\zeta_c = 0$), which is slightly
less than the value given by Li et al. (2015) and provides another indication of the stress dependence of the accumulation
parameter ($\alpha$). Further, Li et al. (2020) confirm the accumulation parameter ($\alpha$) to be independent from cyclic load magnitude
($\zeta_b$) similar to the results presented in the article at hand.

Somehow different are the findings of Klinkvort and Hededal (2013), where the accumulation parameter ($\alpha$) depends on the
cyclic load magnitude ($\zeta_b$) as well as the cyclic load ratio ($\zeta_c$). For cyclic one-way loading ($\zeta_c = 0$) the accumulation parameter
($\alpha$) according to Klinkvort and Hededal (2013) results directly from the equation for $T_{b,K\&H}$ in Table 1, which for example
yields a value of 0.231 for a cyclic load magnitude ($\zeta_b$) of 0.4, linearly further increasing for higher load magnitudes ($\zeta_b$). This
is contrary to the findings of most other authors mentioned in Table 1 except LeBlanc et al. (2010) whose approach is not
directly comparable as it is not based on Eq. 1. In addition, the Klinkvort and Hededal (2013) approach seems to provide very
high accumulation parameters ($\alpha(\zeta_c = 0)$) compared to other methods, at least for load magnitudes ($\zeta_b$) larger than 0.2. Here,
the definition of the reference load $H_{ref}$ for the determination of the load magnitude ($\zeta_b$) according to Klinkvort and Hededal
(2013) has to be kept in mind (see Table 2). Anyway, such high accumulation parameters ($\alpha$) from centrifuge tests are contrary
to the findings of Richards et al. (2021) and the assumption of decreasing accumulation parameters ($\alpha$) with stress-level.
Nevertheless, the results of Klinkvort and Hededal (2013) support the assumption that the accumulation parameter ($\alpha$) is
independent of the soil relative density ($D_r$), which has also been found from the present study, at least for cyclic one-way
loading ($\zeta_c = 0$).

According to the approach of Truong et al. (2019), the accumulation parameter ($\alpha$) for cyclic one-way loading ($\zeta_c = 0$) is
independent from cyclic load magnitude ($\zeta_b$) as already proposed by Li et al. (2020) and also found in this study. Nevertheless,
it linearly decreases with soil relative density ($D_r$). For relative densities ($D_r$) of 0.4 and 0.6, respectively, as used in the
experiments presented above, unidirectional cyclic loading ($\zeta_c = 0$) results in accumulation parameters of 0.212 and 0.168
using the approach of Truong et al. (2019). This is significantly higher than the values obtained in the present study
($\alpha_{mean}(\zeta_c = 0) = 0.1169$) and contradicts the assumption of a decreasing accumulation parameter ($\alpha$) with stress-level in that the
Truong et al. (2019) approach is based on centrifuge tests. On the other hand, this approach yields an accumulation parameter
($\alpha$) of 0.113 for a soil relative density ($D_r$) of 0.85, which is much closer to the value resulting from this study. Possibly, a
stress-dependent conversion of the soil relative density ($D_r$), as proposed by LeBlanc et al. (2010), could provide an explanation
for the resulting deviations (see section 2). However, the dependence of the accumulation parameter ($\alpha$) for unidirectional
loading ($\zeta_c = 0$) on the soil relative density ($D_r$) proposed by Truong et al. (2019) contradicts the results of the present study as
well as those of Klinkvort and Hededal (2013) or Li et al. (2020).

Further, the influence of a variable cyclic load ratio ($\zeta_c$) on the accumulation parameter ($\alpha$) is now discussed. Due to the above
mentioned, partly different dependencies of the estimation approaches for the accumulation parameter ($\alpha$), however, a direct
comparison is not possible. In order to enable a reliable comparison, the results for the accumulation parameter ($\alpha$) with cyclic
load ratio ($\zeta_c$) according to the different approaches presented are shown in normalized form in Fig. 9. By normalizing to
$\alpha(\zeta_c = 0)$, the previously mentioned differences of the approaches with respect to the accumulation parameter ($\alpha$) for one-way
loading ($\zeta_c = 0$) are omitted, so that the influence of the cyclic load ratio ($\zeta_c$) can be considered in isolation. Only the influence
of the soil relative density ($D_r$) according to the approach of Li et al. (2020) cannot be excluded in this way due to the two
proposed nonlinear functions for $T_{c,L}(\zeta_c, D_r)$ (see Table 1). For this reason, Fig. 9 shows two curves for this approach, where
both curves define the limits of applicability of the Li et al. (2020) method with respect to the soil relative density
($0.5 \leq D_r \leq 0.8$). The results according to Peralta (2010) and Li et al. (2015) are not depicted in Fig. 9 as both methods only
propose an accumulation parameter ($\alpha$) for cyclic one-way loading ($\zeta_c = 0$). To allow a comparison with the results of the



present study, also the lower and upper bound curves for the accumulation parameter (α) determined and proposed in section 4.33 are plotted in normalized form in Fig. 9.

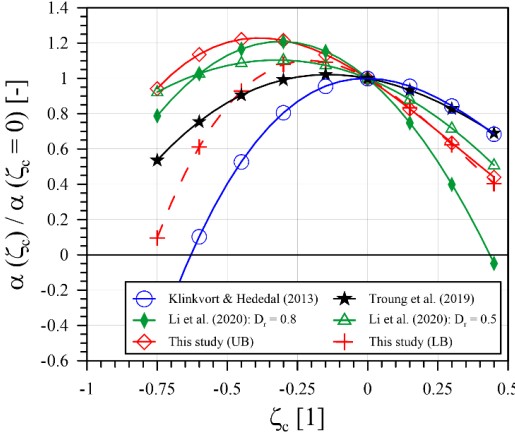

**Figure 9: Normalized accumulation parameter (α) with cyclic load ratio ($\zeta_c$) for different approaches and comparison with the proposed lower bound (LB) and upper bound (UB) curves.**

Fig. 9 shows that in particular the approach according to Li et al. (2020) fits well with the results of the current study. For both soil relative densities ($D_r$) of 0.8 and 0.5, the results are within the proposed limits (LB and UB) for two-way loading ($\zeta_c < 0$), while for one-way loading ($\zeta_c \geq 0$) the Li et al. (2020) curve for a soil relative density ($D_r$) of 0.8 is increasingly divergent and below the proposed boundaries (conservative). The Truong et al. (2019) approach also shows a qualitatively similar shape to the proposed boundary curves, but overall is slightly below the lower bound curve for cyclic two-way loading ($\zeta_c < 0$), and moderately above for one-way loading ($\zeta_c \geq 0$). All the aforementioned curves show a maximum value between approximately 1.23 and 1.02 for an unbalanced two-way loading with cyclic load ratio ($\zeta_c$) in the range of -0.4 to -0.15. This is generally also in agreement with the findings of LeBlanc et al. (2010) who report a maximum accumulation for cyclic two-way loading with a cyclic load ratio ($\zeta_c$) of -0.6, but whose approach is not included in this comparison due to the different formulation of this approach (see Table 1). Somehow different are the findings of Klinkvort and Hededal (2013), whose approach provides a maximum accumulation parameter (α) for cyclic one-way loading with complete unloading in each cycle ($\zeta_c = 0$). For more positive cyclic load ratios ($\zeta_c > 0$), the normalized accumulation parameter according to Klinkvort and Hededal (2013) resembles the values of the Truong et al. (2019) approach that lie slightly above the proposed upper bound curve.

## 6 Conclusions

In this paper, a brief summary of current regulations and recommendations for the serviceability limit state dimensioning of offshore monopile foundations in sand supporting wind turbines was given. Based on this summary, it was shown that current offshore guidelines (DNV GL, 2018 and API, 2014) provide design requirements but do not recommend appropriate design procedures for predicting deformations for large diameter piles subjected to long-term lateral cyclic loading. Instead, a variety of different methods for the prediction of such deformations can be found in literature, some of which have been briefly presented. Based on example calculations, it was shown that the proposed methods for deriving the cyclic load-deformation behaviour of monopile foundations yield partly significantly different results. Furthermore, it could be shown that depending on the chosen approach, the results exhibit a partly contradictory trend with regard to the influence of some input variables such as load or soil parameters. To better understand this outcome, a comprehensive experimental small-scale model test campaign involving approximately 150 single tests on different pile-soil systems subjected to varying loading conditions ($\zeta_b$ and $\zeta_c$) being representative for the environmental conditions of an offshore monopile foundation (Jalbi et al., 2019) has been conducted and evaluated. Based on the results, it could be shown that a power function (Eq. 1) is very suitable for representing



the pile head displacement accumulation of rigid piles under different cyclic one- and two-way loading conditions with constant mean load and amplitude. For the accumulation parameter ($\alpha$) of the power function, it was found from the conducted

tests that it is almost independent of the cyclic load magnitude ($\zeta_b$), the soil relative density ($D_r$), the load eccentricity (h) and the pile embedment length (L) for cyclic one-way loading ($\zeta_c = 0$), as long as the pile-soil systems can be classified to behave rigid. Comparison of these findings and the determined mean value for $\alpha(\zeta_c = 0) = 0.1169$ with values derived from other methods showed that this observation is only shared by some authors. Furthermore, it was shown that the determined absolute value of the accumulation parameter for one-way cyclic loading ($\zeta_c = 0$) seems to exhibit a stress dependence. Therefore, the

direct transfer of the presented results to true scale cannot be recommended. With regard to the influence of different cyclic load ratios ($\zeta_c$), the test results of the test campaign conducted showed a relatively clear trend. Maximum accumulation and therefore accumulations parameters ($\alpha(\zeta_c)$) in general result from unbalanced two-way loading (-0.4 < $\zeta_c$ < -0.15) and lead to an increase of the accumulation parameter ($\alpha$) by a factor of up to 1.23 compared to one-way loading ($\zeta_c = 0$). Since the determined accumulation parameters for variable cyclic load ratios ($\alpha(\zeta_c)$) on the one hand vary slightly due to experimental

scatter, and on the other hand seem to be at least slightly influenced by other variables (e.g. $D_r$), two equations for an upper and a lower bound of the accumulation parameter ($\alpha(\zeta_c)$) were proposed. If the proposed limit curves for the accumulation parameter ($\alpha(\zeta_c)$) are normalized to the accumulation parameter for cyclic one-way loading ($\alpha(\zeta_c = 0)$), then a comparison with the results obtained by other approaches shows relatively good agreement. In order to be able to make a prediction of the cyclic displacement accumulation of a pile using the power function according to Eq. 1, the accumulation parameter for one-way

cyclic loading ($\alpha(\zeta_c = 0)$) should first be known as accurately as possible. This can be achieved by site-specific numerical simulations or centrifuge testing to avoid unwanted stress effects. Another possibility would be to determine functions for the stress dependent conversion of the proposed accumulation parameter ($\alpha$) from the present small-scale model tests (see, e.g. Richards et al., 2020). For deviating load conditions (varying cyclic load ratios ($\zeta_c$)), a range of possible accumulation parameters ($\alpha(\zeta_c)$) can be estimated using the proposed upper and lower bound curves by normalizing these curves and

multiplying the resulting factor with the before determined site-specific accumulation parameter for one-way loading ($\alpha(\zeta_c = 0)$). Further research should especially focus on the accurate determination of the accumulation parameter for cyclic one-way lateral loading ($\alpha(\zeta_c = 0)$).

**Acknowledgement**

This study was carried out in the scope of the research project "Accumulation of lateral displacements of piles under general

cyclic one- and two-way loading" funded by the Deutsche Forschungsgemeinschaft (DFG, German Research Foundation) - project no.: 393683178. The authors sincerely acknowledge DFG support.

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
