# Peer review of "A model test study on the parameters affecting the cyclic lateral response of monopile foundations for offshore wind turbines embedded in non-cohesive soils."

_Wind Energy Science, 2021_

## Author Response (AR1)

**RC1 (Tomas Sabaliauskas):**

A very exciting topic, touching very important questions. At the same time not digging deep, not raising questions that would be radical. Instead, staying within a small box, a small niche of a small paradigm - discussing importance of one parameter between a few very similar, simple "models" that plot curves on top of data. Nothing extremely significant stated, but the careful points made are very well written, presented. Easy to follow. Yet... nothing "shocking" proposed or discovered.

The discussion shows various models compared. All the models are built using almost identical parameter names, assuming nearly identical theory / explanation. They all assume knowing the load and initial density - is enough, and nothing more will ever be needed. Completely ignoring the elephant in the room - non of these "solutions" truly solve irregular loaded structure behavior.

None of them quantify disturbed soil states. None of them show how to "disturb a foundation back to initial state". Real structures go throgh millions of irregular loading cycles, and we have to start modelling this with good precision some day (sooner preferably). Instead, literature tends to be stuck discussing "one parameter" in some few methods none of whitch come even close to the "real" solution. Topic limited to "which method is the least bad - out of all the not good enough".

This topic is ripe for radical, provocative, alternative / opposite, new and unseen, different tests, findings, models and topics... it is simply screaming for it. But students are taught to be nice, polite and not question absurd limitations imposed by convention. Not to take risks. Be calm, polite, simple.. be nice, agree and stay silent - remain not noticed. Confirming, submissive... Do not question superiors. Do not provoke.

Repeat the same tests. Even if same curve is hard to produce twice - fake confidence in the model, say it is good and predicting well. Never do tests - others had not done before you. Copy others. Follow others... stay behind - follow. This paper show that quite strongly. It is too gentle. Proud and protective of methods that do not work well (even if no one has better ones yet). I'd like a mention of the big problem - none of these are "enough". Something "big" is not there - in them all.

I wish the elephant in the room to be mentioned, exposed and attacked very agressively. Disturbed soil, irregular loading cycles - are neither tested or discussed in the paradigm. All "research" is stuck repeating the same "constant amplitude" loads - and then arguing how to fit such basic case. With each case either re-calibrating the model parameters, or changing position of some parameter. Each test - new model, new calibration. Rather than looking for "common pattern" in the "big picture". Rather than raising the question of "how does the physical system work?", "what govern the physical properties, fundamentally?", "how to control the shape, size and position of stiffness hysteresis loop"... instead, the discusiion is "how to compliment experts", "who to like", "who to side with". I see no sides that are "right" yet. The "world best" is not good enough - yet.

Paper does mention conflict. Density having lower influence than expected. Some cyclic accumulation coefficient not having influence in some cases. But that does not address the big problem - irregular cycles. Controlling / predicting shape, size, position of individual hysteresis loop...

I was expecting more, after reading the title. I was expecting either GPU (parallel computing) compatible models or test results bridging across various testing cases. Tests that explore irregular cyclic loading - combining drained and undrained sequences. Examples disturbing a foundation back to initial state, controlling the shape size and position of individual stiffness hysteresis loops - this had been achieved, had been published. It is plausible to demonstrate and observe very easily. Although, not very popular. Not done by famous, high cited "experts".

The experts tend to build larger piles and repeat the same test to reach the same conclusions - by calibrating the same models with the same coefficients... then publish the same books, with the same

text but a few extra pictures of a larger pile. Despite it's apparent aim to expose contradictions - this paper still fits well in that path.

My main criticism for this paper is - it takes no risks. It says nothing new. It talks about other people and others work. Trying to compliment and carefully "fit in with the others". Too careful for my taste. There is nothing provocative or exciting discovered, observed, assumed, tested, etc... no one insulted. No theory "crushed" by the "extreme new empirical observations". Above all it shows a lot of books had been read, and the "important names" are known. No grammar errors made - due to author being excited about the topic "too much". Good for a luxury diner with a stale conversation - less good for real / meaningful scientific progress.

Thus, I strongly recommend to publish this work. It is very good. It is just like the rest of the publications. Does not stand out in any significant way. Does not excite. Feels dry. Simple. Perfect for a safe "yes man" career within academia / teaching / repeating. Fit for a young pentionist - saving for pension in 20's. It contradicts nothing and insults no one, by proposing no "really new" ideas. With perfect grammar, and fluent writing. Totally by the book, following all rules precisely. It is a perfect paper by academic standards. Good for teaching, bad for science.

**AC1 (Dennis Frick):**

The authors would first like to thank the reviewer for the detailed comment. Regarding the mentioned criticism, it can only be stated that it was not the intention of the authors to attack other authors and their publications as well as findings or to provoke them. Rather, this paper is intended to be a factual and clear compilation of the current state of the art as well as already existing (simple) methods for the prediction of pile deformations under cyclic lateral loading.

The presented results of the extensive test campaign shall help to evaluate and at best to confirm the already existing methods. Even though comparable experiments have already been carried out by many other scientists and the results have been published, the authors of this article are not aware of any publication in which such a number of different system parameters have been systematically investigated, which could possibly make the reported results of the experiments interesting for others.

Furthermore, the authors are of course aware that such simple methods as those mentioned or developed in this article can only provide a rough and first estimation of pile deformations under cyclic constant loads. Considering the complexity of this field of research, the loading conditions in situ and the lack of other accepted methods or standards that precisely regulate the prediction of deformations for large diameter piles, such simple and engineering methods are anyway well suited at least for a first preliminary design.

However, the authors agree with the reviewer that in the medium or long term, it would be desirable to develop new (numerical) prediction methods or models that can represent the complex behaviour of cyclically laterally loaded piles much more accurately and in its entirety. To get closer to this objective, also the publication of results of carefully planned and performed experimental tests, which can for example be used for the calibration or validation of numerical models, is a small contribution.

Finally, it should be noted that no changes have been made to the present article in accordance with Reviewer 1's comments.

**RC2 (Gudmund Reidar Eiksund):**

The paper can be excepted as it is with a small modification. Some of the test results are covered by the legend in figure 6.

The paper presetns a model for estimating accumulated pile displacement due to cylclic loading. The influence of average and cylic loading component are well covered, but a dicussion that is missing is the influence of variable load apmplitued during the life time. How can multiple storm events be represented by an equivalent number of oad repetitions in this model?

**AC2 (Dennis Frick):**

The authors would also like to thank the second reviewer for his time and commentary.

The authors are aware that, at least for test series 8, the legend in figure 6 partially obscures the results shown. This could be changed by altering the scaling of the y-axis of the corresponding partial figure. However, in order to ensure a better comparability of the presented test results among each other, the authors decided to choose a uniform scaling for all test series depicted in Figure 6. It should also be noted that the results of test series 8 in particular were classified as non-representative in the text, so that the overlapping of the results by the legend is not disturbing from the authors' point of view.

Due to the reasons mentioned above, no changes have been made to Figure 6 by the authors. Should the reviewer or the editor wish to change Figure 6 anyway, then the authors will of course implement and incorporate the above-mentioned adjustments.

With regard to the comment on the discussion of the influence of variable loads and how these can be taken into account, a short paragraph has now been added to chapter 2 "State of the art" (lines 128 to 136). However, a detailed presentation of this problem was intentionally omitted in the context of this manuscript, since it would be another very broad topic, which has already been discussed in many other publications (two of them have been added as a reference, see new paragraph). Due to this reason, the focus of the present paper should be only on constant amplitude cyclic loads and the load as well as system parameters influencing the cyclic load-displacement behaviour of piles.